# Taught Well Learned Ill: Towards Distillation-conditional Backdoor Attack

**Yukun Chen**[1,2,*], **Boheng Li**[3,*], **Yu Yuan**[1,2,*], **Leyi Qi**[1,2],
**Yiming Li**[3,✉], **Tianwei Zhang**[3], **Zhan Qin**[1,2], **Kui Ren**[1,2]
[1]State Key Laboratory of Blockchain and Data Security, Zhejiang University
[2]Hangzhou High-Tech Zone (Binjiang) Institute of Blockchain and Data Security
[3]Nanyang Technological University
{yukunchen, qinzhan, kuiren}@zju.edu.cn; BOHENG001@e.ntu.edu.sg;
{yuyuan21cath, liyiming.tech}@gmail.com; leyi-qi@outlook.com;
tianwei.zhang@ntu.edu.sg

## Abstract

Knowledge distillation (KD) is a vital technique for deploying deep neural networks (DNNs) on resource-constrained devices by transferring knowledge from large teacher models to lightweight student models. While teacher models from third-party platforms may undergo security verification (*e.g.*, backdoor detection), we uncover a novel and critical threat: distillation-conditional backdoor attacks (DCBAs). DCBA injects dormant and undetectable backdoors into teacher models, which become activated in student models via the KD process, even with clean distillation datasets. While the direct extension of existing methods is ineffective for DCBA, we implement this attack by formulating it as a bilevel optimization problem and proposing a simple yet effective method (*i.e.*, SCAR). Specifically, the inner optimization simulates the KD process by optimizing a surrogate student model, while the outer optimization leverages outputs from this surrogate to optimize the teacher model for implanting the conditional backdoor. Our SCAR addresses this complex optimization utilizing an implicit differentiation algorithm with a pre-optimized trigger injection function. Extensive experiments across diverse datasets, model architectures, and KD techniques validate the effectiveness of our SCAR and its resistance against existing backdoor detection, highlighting a significant yet previously overlooked vulnerability in the KD process. Our code is available at `https://github.com/WhitolfChen/SCAR`.

## 1 Introduction

Deep Neural Networks (DNNs) have achieved excellent performance, leading to their widespread adoption in numerous safety-critical domains [1, 34, 42]. To achieve better performance, DNNs are typically designed to be deeper and wider [19, 29]. However, due to limitations of computational and memory resources, such heavy models are clumsy to deploy on resource-constrained devices (*e.g.*, IoT devices). Knowledge distillation (KD) [24, 59, 83], a technique that enhances the performance of lightweight models (student) by transferring knowledge from larger models (teacher), has gained increasing popularity. With fewer training steps and less data, KD enables small models to achieve accuracy and generalization performance comparable to large models [56]. Therefore, an increasing number of users obtain powerful yet cumbersome large models from third-party platforms (*e.g.*, GitHub [69] and Hugging Face [36]) to serve as teachers for training lightweight student models.

---

*The first three authors contributed equally to this work. ✉ Corresponding author: Yiming Li.

39th Conference on Neural Information Processing Systems (NeurIPS 2025).

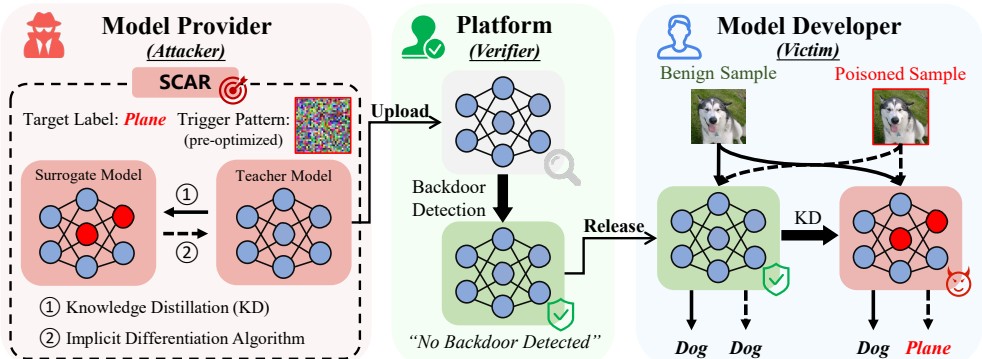

Figure 1: The attack scenario of our SCAR. A malicious model provider (*i.e.*, attacker) leverages SCAR to implant a dormant backdoor into the model, which behaves normally even when fed with poisoned inputs. The compromised model is then uploaded to a third-party platform (*i.e.*, verifier) for backdoor detection. Upon passing the security check, the model is released to model developers (*i.e.*, victim). When the developer utilizes the original model for inference, it performs as expected. However, once it undergoes further development via KD (with benign samples), inputs with the attacker-specified trigger can activate the backdoor in the student model, leading to misclassification.

However, such third-party models may introduce security vulnerabilities to malicious attacks. Among these, backdoor attacks, which implant hidden malicious behavior during model training [16, 21, 46], pose an especially significant concern. A backdoor-compromised model operates as expected under normal conditions but exhibits attacker-specified malicious behavior when a specific trigger condition is satisfied [27, 61, 78]. To ensure the security of third-party models, platforms like Hugging Face are increasingly implementing security validation for models uploaded by providers, with backdoor detection serving as a crucial component [13, 37]. Once a model passes this validation, it is often deemed secure and released to developers for further development. Developers might subsequently employ these 'safety-verified' models as teachers to guide the training of lightweight student models via KD. This practice, however, introduces a compelling research question: *If the teacher model is verified as 'backdoor-free' and the distillation dataset is clean, can we truly guarantee that the resulting student model is also free from backdoor threats?*

Unfortunately, the answer to the above question is negative, although executing such backdoor attacks on the student model is not trivial. In this paper, we explore the potential threat of distillation-conditional backdoor attacks, where the backdoor remains dormant and undetectable in the teacher model but becomes activated in the student model through KD, as shown in Figure 1. Arguably, the most straightforward method to design these attacks is to extend the anti-distillation backdoor attack (ADBA) [23], whose backdoor could be preserved during the KD process. In general, this extension involves fine-tuning to slightly 'mask' the backdoor within a model compromised by ADBA (dubbed 'ADBA (FT)') to make the logit of the ground-truth label slightly higher (instead of significantly lower) than that of the attacker-specified target label when processing poisoned inputs. This manipulation simultaneously conceals the model's backdoor behavior while positioning its decision boundary for poisoned inputs near a vulnerable tipping point. Intuitively, KD may cause a shift in the student model's decision boundary relative to that of the teacher model, thereby reactivating the masked backdoor. However, we find that this method often fails to implant the backdoor into student models. We argue that this is probably because the fine-tuning process masks the teacher's backdoor solely based on its own behavior, lacking guidance derived from the dynamic KD process.

Motivated by aforementioned findings and understandings, we formulate the compromised model training process as a bilevel optimization problem and propose **SCAR** (Stealthy distillation-Conditional bAckdooR attack) to inject distillation-conditional backdoors into teacher models. In general, we derive a surrogate model via finite KD optimization steps and optimize the teacher model utilizing outputs from the surrogate. Specifically, in this bilevel problem, the inner optimization minimizes the output prediction discrepancy between the surrogate (student) and the teacher (simulating the KD process), while the outer optimization maximizes the attack performance of poisoned samples against the surrogate, without compromising the teacher's accuracy and robustness. To address the challenge, where the teacher's gradient cannot be computed directly due to the inner loop and the surrogate being a separate entity, we derive an implicit differentiation algorithm [25] to capture the interdependence between the inner and outer optimization. We approximate the teacher's gradient utilizing reverse-mode automatic differentiation with a finite number of fixed-point itera-

tions. In particular, we pre-optimize the trigger pattern to further simplify the optimization of the above bilevel problem and reduce the parameter search space. Our attack alerts developers to notice a potential false sense of security or consensus that distilling a student model based on a 'backdoor-free' teacher model with benign samples is always safe. As such, developers should always detect the distilled student model, no matter whether the teacher model is regarded as 'secure'.

In summary, our main contributions are three-fold: **(1)** We introduce a novel backdoor attack paradigm, $i.e.$, distillation-conditional backdoor attack (DCBA), where dormant backdoors in the teacher model can be activated in the student model via the KD process even with benign samples. **(2)** We reveal that the direct extension of existing methods ($e.g.$, ADBA (FT)) is ineffective as a DCBA and its potential reasons. Based on these understandings, we formulate the DCBA as a bilevel optimization problem and propose a simple yet effective method ($i.e.$, SCAR). Our SCAR derives an implicit differentiation algorithm and leverages a pre-optimized trigger pattern to solve this problem. **(3)** We conduct extensive experiments across multiple datasets, model architectures, and KD techniques (simulating diverse developer behaviors), demonstrating the effectiveness of SCAR and its resistance to potential backdoor detection. Our work highlights the urgent need to always detect the distilled student model, no matter whether the teacher model is deemed secure.

## 2 Background

### 2.1 Knowledge Distillation

Knowledge distillation (KD) [3, 30, 35] aims to transfer knowledge from a powerful yet cumbersome teacher model to a lightweight student model, enabling performant deployment of DNNs on resource-limited devices ($e.g.$, IoT devices). Typically, the training objective for the student model involves minimizing a composite loss function defined as: $\mathcal{L}_s = \mathcal{L}_{CE} + \delta \cdot \mathcal{L}_{KD}$. In this formulation, $\mathcal{L}_{CE}$ is the cross-entropy loss between the student's output logits and the ground-truth labels, and $\mathcal{L}_{KD}$ denotes the distillation loss that encourages the student model to mimic the output distribution or intermediate features imparted by the teacher model. $\delta$ is a balancing coefficient.

Depending on how knowledge is extracted and transferred via $\mathcal{L}_{KD}$, existing KD methods can be classified into three types [24, 59, 83]: **(1)** *Response-based KD* [30], which directly aligns the output distribution ($e.g.$, logits or probabilities) of student models with that of teacher models, often utilizing metrics like Kullback-Leibler (KL) divergence [30]. **(2)** *Feature-based KD* [3], focusing on matching intermediate representations between the student and teacher models, such as feature maps [3, 68] or attention maps [87]. **(3)** *Relation-based KD* [35], which transfers structural knowledge from teacher models to student models by capturing and aligning relationships, including relational information between different data points [35, 64] or features [73].

### 2.2 Backdoor Attack

Backdoor attacks [46] aim to embed hidden malicious behaviors into DNNs during training, typically by poisoning a subset of the training data with predefined trigger patterns. The compromised model behaves normally on benign inputs but exhibit attacker-specified behavior when a particular trigger condition is met. The earliest backdoor attack can be traced back to BadNets [27], which injects backdoors by adding a small white patch to a subset of the training samples and changing their labels to a target class. Subsequent work has explored various backdoor attack methods, including the design of invisible [5, 60] or diversified [75, 90] triggers, attacks effective in physical-world scenarios [11, 79], and those targeting specific tasks [2, 54], among other advancements [17, 22, 77].

**Backdoor Attacks against Knowledge Distillation.** Since the student model typically learns only the benign behavior of the teacher from a clean distillation dataset, most existing backdoors struggle to survive the KD process [86]. Currently, some studies [4, 7, 23] have focused on designing distillation-resistant backdoor attacks. Anti-Distillation Backdoor Attack (ADBA) [23] is the first to introduce the idea of training a distillation-resistant backdoor in the teacher model by leveraging a shadow model to simulate the KD process. Recently, a backdoor attack targeting feature-based KD is proposed, which encodes the backdoor knowledge into specific neuron activation layers [4]. In addition, backdoors have also been shown to potentially survive the KD process in large language models [7]. However, to the best of our knowledge, it remains unexplored whether distilling a (seemingly) clean model using a benign dataset could still result in backdoored student models.

## 2.3 Backdoor Detection

To determine whether a model contains backdoor threats, researchers have proposed various detection methods, which can be broadly categorized into two types: **(1)** *Trigger inversion-based methods*, which aim to reverse-engineer potential attacker-specified backdoor triggers. Neural Cleanse (NC) [74], one of the earliest methods, attempts to reconstruct such triggers through optimization techniques and identifies backdoors by analyzing properties of the recovered triggers, such as whether their size is abnormally small. Most recently, BTI-DBF [82] enhances trigger inversion by decoupling benign features and simultaneously minimizing the discrepancies in benign features while maximizing those in poisoned features. **(2)** *Trigger inversion-free methods*, which detect backdoors without explicitly reconstructing triggers, typically by adopting specific strategies. For example, some methods detected backdoors by examining abnormal behaviors of poisoned samples during inference, such as scaled prediction consistency [28, 32] and concatenation unalignment [85]. More details about related work are in Appendix H.

# 3 Methodology

## 3.1 Threat Model

As illustrated in Figure 1, we consider a three-party scenario in which a malicious model provider (*i.e.*, attacker) uploads a compromised pre-trained model to a third-party platform (*i.e.*, verifier). The platform typically supports backdoor detection but rarely offers mitigation, constrained by limited data and computational resources. Once the model passes detection, it is released to downstream developers (*i.e.*, victims). Although appearing benign, the model carries a dormant backdoor that remains inactive until the developer applies knowledge distillation (KD), at which point the student model misclassifies inputs containing attacker-specified triggers.

**Attacker's Goals.** The attacker aims to train a DNN model with a dormant backdoor, achieving the following three objectives: **(1)** The model achieves correct classification on both benign and poisoned inputs, making the backdoor inactive. **(2)** The model can evade detection by third-party backdoor detection; specifically, existing detection techniques fail to identify the dormant backdoor. **(3)** After undergoing any form of KD to the model, the resulting student model obtains good performance on benign samples but exhibits misclassification on poisoned ones.

**Attacker's Capability.** We primarily focus on the common attack scenario involving pre-trained models [46, 51]. Specifically, the attacker has full control over the training phase of the compromised model, including the training dataset, optimization algorithm, loss function, and hyperparameters. However, the attacker can only provide a pre-trained model, without access to any other information. In particular, they are unaware of which backdoor detection method will be used by the third-party verifier, as well as which specific KD technique will be adopted by the model developer.

## 3.2 An Ineffective Baseline: Anti-Distillation Backdoor Attack with Fine-tuning

Arguably, to achieve the attack goals, we can first train a teacher model with a distillation-resistant backdoor and then fine-tune it to slightly 'mask' the backdoor. Intuitively, this manipulation simultaneously conceals the model's backdoor behavior while positioning its decision boundary for poisoned inputs near a vulnerable tipping point, which can be shifted back to reactivate the masked backdoor during the KD process. Currently, anti-distillation backdoor attack (ADBA) [23] can inject such distillation-resistant backdoors into the teacher model. As such, during fine-tuning, we mask the backdoor in the ADBA-compromised teacher model by adjusting the logits such that, for poisoned inputs, the logit of the ground-truth label is only slightly higher (instead of significantly lower) than that of the attacker-specified target label. Formally, given such a teacher model $\mathcal{F}'_t(\cdot; \boldsymbol{\lambda})$ already, its corresponding trigger inject function $G'(\cdot)$, and a fine-tuning dataset $\mathcal{D}' = \{(\boldsymbol{x}_i, y_i)\}_{i=1}^{M}$, this straightforward baseline (dubbed 'ADBA (FT)') aims to solve the following problem:

$$\min_{\boldsymbol{\lambda}} \frac{1}{M} \sum_{(\boldsymbol{x}_i, y_i) \in \mathcal{D}'} \left[ \mathcal{L}_{CE}(\mathcal{F}'_t(\boldsymbol{x}_i; \boldsymbol{\lambda}), y_i) + \eta \cdot \max\{\mathcal{F}'_t(G'(\boldsymbol{x}_i); \boldsymbol{\lambda})|_{y_t} + k - \mathcal{F}'_t(G'(\boldsymbol{x}_i); \boldsymbol{\lambda})|_{y_i}, 0\} \right],$$

(1)

where $\mathcal{L}_{CE}$ denotes the standard cross-entropy loss, $k$ and $\eta$ are hyper-parameters, $\mathcal{F}'_t(G'(\boldsymbol{x}_j); \boldsymbol{\lambda})|_{y_j}$ and $\mathcal{F}'_t(G'(\boldsymbol{x}_j); \boldsymbol{\lambda})|_{y_t}$ represent the output logits $\mathcal{F}'_t(G'(\boldsymbol{x}_j); \boldsymbol{\lambda})$ corresponding to the ground-truth

and target label, respectively. The first term of the above optimization ensures accurate predictions on benign samples, while the second term enforces that the logit of the ground-truth label is at least $k$ higher than that of the target label. However, we will show that ADBA (FT) has limited effectiveness in attacking student models in many cases. We find that this ineffectiveness is probably due to the fine-tuning relying solely on the teacher's output, without accounting for the dynamic KD process.

### 3.3 The Design of SCAR

Motivated by the above findings and insights, we propose SCAR (Stealthy distillation-Conditional bAckdooR attack), which can effectively leverage information from the KD process. Specifically, we formalize the attack goals as a bilevel optimization problem and develop an appropriate optimization strategy. We also pre-optimize the trigger injection function to ensure a favorable initialization for effectively and efficiently solving the bilevel problem. Besides, we provide a preliminary analysis to explain the effectiveness of our SCAR in Appendix A.

#### 3.3.1 Optimization Objective of SCAR

Here, we formalize the attack goals, which also define the optimization objective of SCAR, as a bilevel optimization problem. Given a training dataset $\mathcal{D} = \{(\boldsymbol{x}_i, y_i)\}_{i=1}^N$, we specify a trigger injection function $G(\cdot)$ and a target label $y_t$ to train a compromised (teacher) model $\mathcal{F}_t(\cdot; \boldsymbol{\lambda})$ with parameter $\boldsymbol{\lambda}$. To simulate the KD process of student models, we introduce a surrogate model $\mathcal{F}_s(\cdot, \boldsymbol{\omega})$, parameterized by $\boldsymbol{\omega}$. Formally, we aim to solve the following bilevel optimization problem:

$$
\begin{aligned}
\min_{\boldsymbol{\lambda}} \mathcal{L}_{out}(\boldsymbol{\omega}(\boldsymbol{\lambda}), \boldsymbol{\lambda}) &\triangleq \frac{1}{N} \sum_{(\boldsymbol{x}_i, y_i) \in \mathcal{D}} \Big[ \mathcal{L}_{CE}(\mathcal{F}_t(\boldsymbol{x}_i; \boldsymbol{\lambda}), y_i) + \alpha \cdot \mathcal{L}_{CE}(\mathcal{F}_t(G(\boldsymbol{x}_i); \boldsymbol{\lambda}), y_i) \\
&\quad + \beta \cdot \mathcal{L}_{CE}(\mathcal{F}_s(\boldsymbol{x}_i; \boldsymbol{\omega}(\boldsymbol{\lambda})), y_i) + \gamma \cdot \mathcal{L}_{CE}(\mathcal{F}_s(G(\boldsymbol{x}_i); \boldsymbol{\omega}(\boldsymbol{\lambda})), y_t) \Big], \\
\text{s.t. } \boldsymbol{\omega}(\boldsymbol{\lambda}) &\in \arg\min_{\boldsymbol{\omega}} \mathcal{L}_{in}(\boldsymbol{\omega}, \boldsymbol{\lambda}) \triangleq \frac{1}{N} \sum_{(\boldsymbol{x}_i, y_i) \in \mathcal{D}} \Big[ \mathcal{L}_{CE}(\mathcal{F}_s(\boldsymbol{x}_i; \boldsymbol{\omega}), y_i) \\
&\quad + \delta \cdot \mathcal{L}_{KD}(\mathcal{F}_s(\boldsymbol{x}_i; \boldsymbol{\omega}), \mathcal{F}_t(\boldsymbol{x}_i; \boldsymbol{\lambda})) \Big],
\end{aligned}
\tag{2}
$$

where $\mathcal{L}_{CE}$ denotes the standard cross-entropy loss, and $\mathcal{L}_{KD}$ represents the knowledge distillation loss, computed as the KL divergence between the output logits of $\mathcal{F}_t$ and $\mathcal{F}_s$. The scalars $\alpha$, $\beta$ and $\gamma$ are temperature coefficients, and $\delta$ is a balancing coefficient. In the outer optimization, the first two terms of $\mathcal{L}_{out}$ reflect the attacker's objective for the teacher model to behave normally on both benign and poisoned samples, while the last two terms aim to ensure that the surrogate model performs normally on benign samples but exhibits backdoor behavior on poisoned ones. The inner optimization is designed to ensure that the surrogate model closely mimics the distillation process of unknown student models. In particular, optimizing the teacher model parameters $\boldsymbol{\lambda}$ requires computing the gradient of $\mathcal{L}_{out}$ with respect to $\boldsymbol{\lambda}$. Since the last two terms of $\mathcal{L}_{out}$ involve $\mathcal{F}_s(\cdot; \boldsymbol{\omega}(\boldsymbol{\lambda}))$ and are thus implicitly dependent on $\boldsymbol{\lambda}$ via $\boldsymbol{\omega}(\boldsymbol{\lambda})$, the gradient computation must account for the dependence of the inner solution $\boldsymbol{\omega}(\boldsymbol{\lambda})$ on $\boldsymbol{\lambda}$. Due to the presence of an inner optimization loop and the fact that the surrogate model is a separate entity, the gradient of $\boldsymbol{\lambda}$ cannot be directly computed via standard backpropagation. To tackle this challenge, we derive an *implicit differentiation algorithm* to approximate the gradients arising from these two terms, effectively capturing the $\boldsymbol{\omega}$-$\boldsymbol{\lambda}$ interdependence. Its technical details are in the following subsection.

#### 3.3.2 Optimization Strategy of SCAR

We hereby derive an *implicit differentiation algorithm* to estimate the gradient $\nabla_{\boldsymbol{\lambda}} \mathcal{L}_{out}$ with respect to the parameters $\boldsymbol{\lambda}$ of $\mathcal{F}_t$. The overall training process of our SCAR is outlined in Algorithm 1. In the following, we provide a detailed explanation of how to compute this gradient.

**Finite Inner Optimization Updates.** Intuitively, a key step in computing $\nabla_{\boldsymbol{\lambda}} \mathcal{L}_{out}$ is to identify $\boldsymbol{\omega}^*(\boldsymbol{\lambda})$, the solution to the inner optimization problem:

$$
\boldsymbol{\omega}^*(\boldsymbol{\lambda}) = \arg\min_{\boldsymbol{\omega}} \mathcal{L}_{in}(\boldsymbol{\omega}, \boldsymbol{\lambda}). \tag{3}
$$

Let $\boldsymbol{\omega}^*(\boldsymbol{\lambda})$ be a suboptimal solution which satisfies the first-order optimality condition:

$$
\nabla_{\boldsymbol{\omega}} \mathcal{L}_{in}(\boldsymbol{\omega}^*(\boldsymbol{\lambda}), \boldsymbol{\lambda}) = \boldsymbol{0}. \tag{4}
$$

**Algorithm 1** SCAR Training Process

---

**Input:** Model $\mathcal{F}_t(\cdot; \boldsymbol{\lambda})$, Surrogate $\mathcal{F}_s(\cdot; \boldsymbol{\omega})$, Trainset $\mathcal{D}$, Trigger function $G(\cdot)$, Target label $y_t$
**Output:** Trained compromised model $\mathcal{F}_t$
**Parameters:** Fix-point iterations $K$, Subset batches $M$, Inner steps $T$, Learning rate $\epsilon$ and $\theta$

1: **for** *each outer optimization epoch* **do**
2:      Reinitialize $\boldsymbol{\omega}_0$;                                             ▷ Initialize inner parameters
3:      **for** $t = 0$ **to** $T-1$ **do**                     ▷ Inner loop: Approximate $\boldsymbol{\omega}^*(\boldsymbol{\lambda})$
4:          Compute $\nabla_{\boldsymbol{\omega}}\mathcal{L}_{in}(\boldsymbol{\omega}_t, \boldsymbol{\lambda})$ with $\mathcal{D}$;
5:          Update $\boldsymbol{\omega}_{t+1} \leftarrow \boldsymbol{\omega}_t - \epsilon \cdot \nabla_{\boldsymbol{\omega}}\mathcal{L}_{in}(\boldsymbol{\omega}_t, \boldsymbol{\lambda})$;            ▷ Eq. (9)
6:      Select subset $\mathcal{D}_s$ ($M$ batches from $\mathcal{D}$) for outer gradient estimation;
7:      Compute $\mathbf{g}_{\boldsymbol{\omega}} \leftarrow \nabla_{\boldsymbol{\omega}}\mathcal{L}_{out}(\boldsymbol{\omega}^*, \boldsymbol{\lambda})$ and $\mathbf{g}_{\boldsymbol{\lambda}} \leftarrow \nabla_{\boldsymbol{\lambda}}\mathcal{L}_{out}(\boldsymbol{\omega}^*, \boldsymbol{\lambda})$ with $\mathcal{D}_s$, $G$ and $y_t$;
8:      Initialize $\mathbf{v}_0 \leftarrow \mathbf{0}$;
9:      **for** $n = 0$ **to** $K-1$ **do**                                         ▷ Eq. (11)
10:          Compute $\mathbf{v}_{n+1} \leftarrow \mathbf{J}_{\Phi,\boldsymbol{\omega}}\mathbf{v}_n + \mathbf{g}_{\boldsymbol{\omega}}$;
11:      Compute approximate gradient $\nabla_{\boldsymbol{\lambda}}\mathcal{L}_{out} \approx \mathbf{g}_{\boldsymbol{\lambda}} + \mathbf{J}_{\Phi,\boldsymbol{\lambda}}^T\mathbf{v}_K$;        ▷ Eq. (12)
12:      Update $\boldsymbol{\lambda} \leftarrow \boldsymbol{\lambda} - \theta \cdot \nabla_{\boldsymbol{\lambda}}\mathcal{L}_{out}$;          ▷ Optimize outer parameters $\boldsymbol{\lambda}$ of $\mathcal{F}_t$
13: **return** $\mathcal{F}_t$

---

In practice, obtaining the exact $\boldsymbol{\omega}^*(\boldsymbol{\lambda})$ is often infeasible. We approximate it by performing a finite number of optimization steps (*e.g.*, $T$ steps of gradient descent) on $\mathcal{L}_{in}$ with respect to $\boldsymbol{\omega}$, typically starting from a randomly reinitialized $\boldsymbol{\omega}$ in each outer epoch. We denote the resulting approximation also by $\boldsymbol{\omega}^*(\boldsymbol{\lambda})$ for simplicity in the following derivation.

**Derivation of Implicit Differentiation.** Given the obtained $\boldsymbol{\omega}^*(\boldsymbol{\lambda})$, using the chain rule, the total gradient of the outer loss with respect to $\boldsymbol{\lambda}$ can be derived as follows:

$$\nabla_{\boldsymbol{\lambda}}\mathcal{L}_{out}(\boldsymbol{\omega}^*(\boldsymbol{\lambda}), \boldsymbol{\lambda}) = \left(\frac{\partial\boldsymbol{\omega}^*(\boldsymbol{\lambda})}{\partial\boldsymbol{\lambda}}\right)^T \nabla_{\boldsymbol{\omega}}\mathcal{L}_{out}(\boldsymbol{\omega}^*(\boldsymbol{\lambda}), \boldsymbol{\lambda}) + \nabla_{\boldsymbol{\lambda}}\mathcal{L}_{out}(\boldsymbol{\omega}^*(\boldsymbol{\lambda}), \boldsymbol{\lambda})|_{direct}. \quad (5)$$

Let $\mathbf{g}_{\boldsymbol{\omega}} \triangleq \nabla_{\boldsymbol{\omega}}\mathcal{L}_{out}(\boldsymbol{\omega}^*(\boldsymbol{\lambda}), \boldsymbol{\lambda})$ and $\mathbf{g}_{\boldsymbol{\lambda}} \triangleq \nabla_{\boldsymbol{\lambda}}\mathcal{L}_{out}(\boldsymbol{\omega}^*(\boldsymbol{\lambda}), \boldsymbol{\lambda})|_{direct}$. Both $\mathbf{g}_{\boldsymbol{\omega}}$ and $\mathbf{g}_{\boldsymbol{\lambda}}$ can be computed directly with standard backpropagation. To compute the Jacobian term $\partial\boldsymbol{\omega}^*/\partial\boldsymbol{\lambda}$, we apply the Implicit Function Theorem by differentiating the optimality condition (4) with respect to $\boldsymbol{\lambda}$:

$$\nabla_{\boldsymbol{\omega}\boldsymbol{\omega}}^2\mathcal{L}_{in} \cdot \frac{\partial\boldsymbol{\omega}^*}{\partial\boldsymbol{\lambda}} + \nabla_{\boldsymbol{\omega}\boldsymbol{\lambda}}^2\mathcal{L}_{in} = \mathbf{0}. \quad (6)$$

Let $H_{\omega\omega} \triangleq \nabla_{\boldsymbol{\omega}\boldsymbol{\omega}}^2\mathcal{L}_{in}$ be the Hessian matrix with respect to $\boldsymbol{\omega}$, and $H_{\omega\lambda} \triangleq \nabla_{\boldsymbol{\omega}\boldsymbol{\lambda}}^2\mathcal{L}_{in}$ be the mixed partial derivative matrix. Assuming $H_{\omega\omega}$ is invertible (guaranteed by strict convexity of $\mathcal{L}_{in}$), we can solve for the Jacobian as follows:

$$\frac{\partial\boldsymbol{\omega}^*}{\partial\boldsymbol{\lambda}} = -(H_{\omega\omega})^{-1}H_{\omega\lambda}. \quad (7)$$

Substituting Eq. (7) into Eq. (5), we have:

$$\nabla_{\boldsymbol{\lambda}}\mathcal{L}_{out}(\boldsymbol{\omega}^*(\boldsymbol{\lambda}), \boldsymbol{\lambda}) = \mathbf{g}_{\boldsymbol{\lambda}} - H_{\omega\lambda}^T H_{\omega\omega}^{-1}\mathbf{g}_{\boldsymbol{\omega}}. \quad (8)$$

**Approximation via Fix-point Iterations.** Directly computing and inverting $H_{\omega\omega}$ is computationally prohibitive for DNNs. Accordingly, we employ an approximation based on reverse-mode automatic differentiation to implicitly compute the vector-Hessian-inverse product. Consider the inner optimization utilizing $T$ gradient descent steps:

$$\boldsymbol{\omega}_{t+1} = \Phi(\boldsymbol{\omega}_t, \boldsymbol{\lambda}) \triangleq \boldsymbol{\omega}_t - \epsilon \cdot \nabla_{\boldsymbol{\omega}}\mathcal{L}_{in}(\boldsymbol{\omega}_t, \boldsymbol{\lambda}), \quad t = 0, \dots, T-1, \quad (9)$$

where $\epsilon$ is the inner loop learning rate, and $\boldsymbol{\omega}_T = \boldsymbol{\omega}^*(\boldsymbol{\lambda})$. Differentiating the fixed-point equation $\boldsymbol{\omega}^* = \Phi(\boldsymbol{\omega}^*, \boldsymbol{\lambda})$ yields $\partial\boldsymbol{\omega}^*/\partial\boldsymbol{\lambda} = (I - \mathbf{J}_{\Phi,\omega})^{-1}\mathbf{J}_{\Phi,\lambda}$, where $\mathbf{J}_{\Phi,\omega} \triangleq \partial\Phi/\partial\boldsymbol{\omega}$ and $\mathbf{J}_{\Phi,\lambda} \triangleq \partial\Phi/\partial\boldsymbol{\lambda}$ are Jacobians evaluated at $\boldsymbol{\omega}^*$. Substituting this into (5) gives:

$$\nabla_{\boldsymbol{\lambda}}\mathcal{L}_{out} = \mathbf{g}_{\boldsymbol{\lambda}} + \mathbf{J}_{\Phi,\lambda}^T(I - \mathbf{J}_{\Phi,\omega})^{-1}\mathbf{g}_{\boldsymbol{\omega}}. \quad (10)$$

Let $\mathbf{v} \triangleq (I - \mathbf{J}_{\Phi,\omega})^{-1}\mathbf{g}_{\omega}$, which can be solved efficiently utilizing an iterative method like Neumann series expansion [57], leading to the iteration:

$$\mathbf{v}_0 = \mathbf{0}, \quad \mathbf{v}_{n+1} = \mathbf{J}_{\Phi,\omega}\mathbf{v}_n + \mathbf{g}_{\omega}, \quad n = 0, 1, 2, ... \tag{11}$$

This iteration converges to $\mathbf{v}$ if the spectral radius $\rho(\mathbf{J}_{\Phi,\omega}) < 1$ [26, 52]. In practice, we truncate this iteration after finite $K$ steps. Let $\mathbf{v}_K$ be the approximation after $K$ steps. The approximate outer gradient is then computed efficiently utilizing vector-Jacobian products:

$$\nabla_{\boldsymbol{\lambda}}\mathcal{L}_{out} \approx \mathbf{g}_{\boldsymbol{\lambda}} + \mathbf{J}_{\Phi,\boldsymbol{\lambda}}^T\mathbf{v}_K, \tag{12}$$

which can be leveraged to optimize the parameters $\boldsymbol{\lambda}$ of $\mathcal{F}_t$.

### 3.3.3 Pre-optimizing for Trigger Injection Function

To simplify the optimization of the above bilevel problem, we attempt to pre-optimize the trigger injection function $G(\cdot; \boldsymbol{\mu})$ to reduce the parameter search space. Following prior backdoor-related works [46], we define $G(\boldsymbol{x}; \boldsymbol{\mu}) = \Pi_{[0,n]^d}(\boldsymbol{x} + \boldsymbol{\mu})$, where $\Pi_{[0,n]^d}(\cdot)$ ensures that the poisoned image remains within a valid pixel range. Let $\hat{\mathcal{F}}_t(\cdot)$ be a pretrained benign teacher model and $\hat{\mathcal{F}}_s(\cdot)$ be the student model distilled from it. Then, the training of $G$ aims to minimize the following loss:

$$\min_{\boldsymbol{\mu}} \sum_{(\boldsymbol{x}_i, y_i) \in \mathcal{D}} \mathcal{L}_{CE}(\hat{\mathcal{F}}_t(G(\boldsymbol{x}_i; \boldsymbol{\mu})), y_t) + \mathcal{L}_{CE}(\hat{\mathcal{F}}_s(G(\boldsymbol{x}_i; \boldsymbol{\mu})), y_t), \quad \text{s.t.} \ \|\boldsymbol{\mu}\|_{\infty} \leq \epsilon_0, \tag{13}$$

where $y_t$ denotes the target label. In general, this pretraining process aims to optimize a natural backdoor trigger pattern $\boldsymbol{\mu}$ that can survive the knowledge distillation process, thereby providing a favorable initialization for the subsequent bilevel optimization. We validate the necessity of pre-optimizing the trigger injection function in our subsequent ablation studies.

## 4 Experiments

In this section, we evaluate the effectiveness of our SCAR across various datasets, model architectures, and knowledge distillation (KD) methods. We then conduct an ablation study and evaluate the resistance to potential backdoor detection methods. In addition, the analysis of the overhead of SCAR is in Appendix G and the visualization of SCAR is provided in Appendix I.

### 4.1 Main Settings

**Datasets, Models, and KD Methods.** We conduct experiments on two classical benchmark datasets, including CIFAR-10 [40] and (a subset of) ImageNet [14] containing 50 classes. We utilize relatively large models such as ResNet-50 [29], VGG-19 [71], and ViT [19] as the compromised teacher models (with ResNet-18 [29] utilized as the surrogate model), and distill each of them into lightweight student models including MobileNet-V2 [70], ShuffleNet-V2 [55], and EfficientViT [50] to validate the effectiveness of SCAR. During KD, we adopt three different types of methods: (1) Response-based KD [30], (2) Feature-based KD [3], and (3) Relation-based KD [35]. Note that our goal is to evaluate the effectiveness of our attack method rather than to train a SOTA model. Therefore, the benign accuracies of our models may be lower than those of SOTA counterparts.

**Attack Setup.** Note that since our SCAR is the first conditional backdoor attack designed specifically for the knowledge distillation setting, we can only compare its performance with a straightforward baseline ADBA (FT) introduced in Section 3.2. For reference, we also report the results of benign models trained without any attack (dubbed 'Benign'). More attack details are in Appendix F.

**Evaluation Metric.** We evaluate the attack performance of different methods based on the backdoor attack success rate (ASR), defined as the accuracy on poisoned samples, for both the teacher and student models. Specifically, for the teacher model, we aim for a dormant and inactive backdoor, which is reflected by a low ASR; whereas for the student model, we seek a strong backdoor effect, indicated by a high ASR. An effective distillation-conditional backdoor attack is indicated by a lower ASR on the teacher model and a higher ASR on the student models.

Table 1: The attack performance (%) on CIFAR-10 and ImageNet. For each case, the best results are **boldfaced**, while all failed cases (student ASR $< 50\%$) are marked in red.

| Dataset | KD Method | Model | ResNet-50 (Teacher) | | MobileNet-V2 (Student A) | | ShuffleNet-V2 (Student B) | | EfficientViT (Student C) | |
| --- | --- | --- | --- | --- | --- | --- | --- | --- | --- | --- |
| | | Attack | ACC | ASR↓ | ACC | ASR↑ | ACC | ASR↑ | ACC | ASR↑ |
| CIFAR-10 | Response | Benign | 94.12 | 0 | 91.92 | 0 | 89.76 | 0 | 86.86 | 0 |
| | | ADBA (FT) | 90.58 | 6.88 | 91.07 | 92.87 | 85.86 | 81.02 | 86.88 | 30.58 |
| | | **SCAR** | 92.47 | 1.50 | 91.62 | **99.94** | 89.15 | **99.02** | 86.82 | **86.31** |
| | Feature | Benign | 94.12 | 0 | 90.92 | 0 | 89.73 | 0 | 86.92 | 0 |
| | | ADBA (FT) | 90.58 | 6.88 | 90.87 | 98.47 | 85.45 | 49.28 | 86.70 | 31.22 |
| | | **SCAR** | 92.47 | 1.50 | 91.01 | **99.90** | 88.48 | **98.22** | 87.74 | **77.28** |
| | Relation | Benign | 94.12 | 0 | 91.77 | 0 | 89.54 | 0 | 86.88 | 0 |
| | | ADBA (FT) | 90.58 | 6.88 | 91.18 | 98.66 | 85.45 | 71.02 | 86.74 | 34.78 |
| | | **SCAR** | 92.47 | 1.50 | 91.29 | **99.93** | 88.25 | **98.44** | 85.78 | **90.09** |
| ImageNet | Response | Benign | 70.08 | 0 | 70.36 | 0 | 65.00 | 0 | 60.32 | 0 |
| | | ADBA (FT) | 61.56 | 2.53 | 61.00 | 45.39 | 60.48 | 37.51 | 56.16 | 13.31 |
| | | **SCAR** | 64.28 | 2.12 | 63.80 | **81.69** | 63.12 | **72.86** | 60.00 | **53.55** |
| | Feature | Benign | 70.08 | 0 | 69.48 | 0 | 66.32 | 0 | 60.44 | 0 |
| | | ADBA (FT) | 61.56 | 2.53 | 61.16 | 37.92 | 60.60 | 24.57 | 59.04 | 36.20 |
| | | **SCAR** | 64.28 | 2.12 | 64.32 | **74.29** | 62.04 | **57.63** | 57.04 | **52.98** |
| | Relation | Benign | 70.08 | 0 | 70.48 | 0 | 63.52 | 0 | 56.80 | 0 |
| | | ADBA (FT) | 61.56 | 2.53 | 61.80 | 42.61 | 61.36 | 20.08 | 55.72 | 19.22 |
| | | **SCAR** | 64.28 | 2.12 | 63.28 | **91.96** | 64.00 | **62.61** | 58.48 | **61.18** |

Table 2: The attack performance (%) of SCAR with/without $\mathcal{F}_s$ or $G$ on CIFAR-10. For each case, the best results are **boldfaced**, while all failed cases (student ASR $< 50\%$) are marked in red.

| KD Method | Model | ResNet-50 (Teacher) | | MobileNet-V2 (Student A) | | ShuffleNet-V2 (Student B) | | EfficientViT (Student C) | |
| --- | --- | --- | --- | --- | --- | --- | --- | --- | --- |
| | Attack | ACC | ASR↓ | ACC | ASR↑ | ACC | ASR↑ | ACC | ASR↑ |
| Response | w/o $\mathcal{F}_s$ | 89.82 | 2.61 | 88.15 | 82.92 | 87.19 | 51.58 | 87.76 | 31.42 |
| | w/o $G$ | 93.81 | 0.72 | 91.47 | 1.03 | 89.47 | 1.06 | 86.05 | 2.09 |
| | **SCAR** | 92.47 | 1.50 | 91.62 | **99.94** | 89.15 | **99.02** | 86.82 | **86.31** |
| Feature | w/o $\mathcal{F}_s$ | 89.82 | 2.61 | 87.94 | 83.68 | 88.04 | 43.11 | 86.11 | 5.29 |
| | w/o $G$ | 93.81 | 0.72 | 91.49 | 1.48 | 88.91 | 1.44 | 87.69 | 1.33 |
| | **SCAR** | 92.47 | 1.50 | 91.01 | **99.90** | 88.48 | **98.22** | 87.74 | **77.28** |
| Relation | w/o $\mathcal{F}_s$ | 89.82 | 2.61 | 87.66 | 80.13 | 89.22 | 81.28 | 86.48 | 54.19 |
| | w/o $G$ | 93.81 | 0.72 | 91.22 | 1.38 | 88.92 | 1.50 | 86.78 | 1.34 |
| | **SCAR** | 92.47 | 1.50 | 91.29 | **99.93** | 88.25 | **98.44** | 85.78 | **90.09** |

## 4.2 Main Results

As shown in Table 1, compared to the baseline attack method, SCAR achieves a higher attack success rate (ASR) on student models while maintaining a low ASR on the teacher model. Specifically, although ADBA (FT) effectively reduces the ASR on the teacher model (ASR $< 10\%$), its effectiveness in attacking student models is limited. For instance, on the CIFAR-10, ADBA (FT) struggles to implant the backdoor into the student model EfficientViT, likely due to substantial architectural differences from the teacher model ResNet-50. On the ImageNet, ADBA (FT) fails to perform the attack altogether. In contrast, SCAR successfully injects backdoors into student models under all three knowledge distillation methods on the CIFAR-10. While its performance declines on the ImageNet, the ASR remains within an acceptable range. This degradation may be attributed to the increased image size in ImageNet, which makes the convergence of the bilevel optimization problem more challenging. Results for the VGG-19 and ViT models are provided in the Appendix B.

## 4.3 Ablation Study

Our method comprises two key components: (**1**) optimization involving the surrogate model $\mathcal{F}_s$, and (**2**) a pre-optimized trigger injection function $G$. To evaluate the effectiveness of each component separately, we conduct an ablation study using ResNet-50 attacked on CIFAR-10. Specifically, to assess the contribution of $\mathcal{F}_s$, we directly fine-tune the teacher model using only $G$, removing the

Table 3: The backdoor detection performance of NC on teacher models compromised by SCAR. Class 0 is the attack-defined target label. False detections are marked in red.

| Model | Anomaly Index of Each Class ($> 2$ indicates a potential backdoor) | | | | | | | | | |
|---|---|---|---|---|---|---|---|---|---|---|
| | **Class 0** | Class 1 | Class 2 | Class 3 | Class 4 | Class 5 | Class 6 | Class 7 | Class 8 | Class 9 |
| ResNet-50 | 0. | 0. | 1.09 | 0.50 | 0.02 | 0. | 1.15 | 0. | 0. | 0.85 |
| VGG-19 | 0. | 0. | 0. | 1.09 | 0. | 0.07 | 2.01 | 0.91 | 0. | 2.10 |
| ViT | 0. | 0. | 0. | 0. | 1.02 | 0. | 0.98 | 0.73 | 0.70 | 0.81 |

involvement of $\mathcal{F}_s$ entirely. To evaluate the necessity of $G$, we replace it with a fixed white patch located at the bottom-right corner of the image as the trigger pattern. We further perform addtional ablation studies on the effects of the surrogate model architecture, the distillation loss weight $\delta$, the number of inner steps $T$, the bilevel optimization hyperparameters ($\alpha$, $\beta$, and $\gamma$), and diverse triggers substituting for $G$, as illustrated in Appendix E.

As shown in Table 2, we evaluate the attack performance of SCAR without the surrogate model (w/o $\mathcal{F}_s$) or without the pre-optimized trigger injection function (w/o $G$). Experimental results indicate that, without $\mathcal{F}_s$, the effectiveness of SCAR significantly degrades or even fails. This is because the teacher model is trained without guidance from the dynamic KD process, leading to a deviation in the optimization direction. Additionally, without $G$, the teacher model attacked by SCAR fails to transfer backdoor knowledge to student models. This may be attributed to the inherent complexity and instability of the bilevel optimization problem, which makes initialization critically important.

## 4.4 Resistance to Potential Backdoor Detection

In this section, we empirically evaluate the evasion performance of SCAR against potential backdoor detection methods. We focus on a scenario where a third-party trusted verifier employs such methods to identify backdoors within the teacher model. To mimic this verification process, we adopt two representative backdoor detection techniques, Neural Cleanse [74] and SCALE-UP [28], to assess the stealthiness of SCAR-injected backdoors in teacher models trained on CIFAR-10. The results demonstrating the evasiveness of SCAR against more advanced detection methods are provided in Appendix C, while its robustness against non-detection defenses is reported in Appendix D.

**Neural Cleanse (NC)** is a classical backdoor trigger inversion (BTI)-based detection method. It leverages the backdoor property that even a small perturbation can cause inputs to be misclassified into the target label. NC first computes universal adversarial perturbations (UAP) for each all-to-one misclassification as potential triggers, then quantifies these triggers using the $\ell_1$-norm, and finally applies an anomaly detection to identify the most likely true trigger. If the anomaly index exceeds 2, the model is considered to have a backdoor with $95\%$ confidence; otherwise, it is regarded as clean.

As shown in Table 3, we evaluate the anomaly index corresponding to the UAP for each class. Only indices greater than zero are retained, where a higher anomaly index indicates a smaller $\ell_1$-norm of the UAP and thus a higher likelihood of a backdoor. The results show that for ResNet-50 and ViT, NC fails to detect any backdoor. For VGG-19, NC yields false positives by mistakenly identifying normal labels, Class 6 and Class 9, as target labels. These findings demonstrate that models compromised by SCAR can effectively evade NC detection.

**SCALE-UP** is a black-box, input-level detection method based on the observation that, when all pixel values are amplified, poisoned samples exhibit significantly more consistent predictions compared to benign ones. It identifies and filters poisoned samples by analyzing the prediction consistency of inputs during the pixel amplification process. We detect the presence of backdoors in the teacher model by observing whether poisoned samples exhibit the prediction consistency behavior.

As shown in Figure 2, both benign and poisoned samples exhibit similar scaled prediction inconsistency across different model architectures compromised by SCAR. Specifically, for conventional backdoor attacks, the prediction confidences on poisoned samples are expected to be more stable than those on benign samples as the amplification times increase, meaning the red curve should consistently lie above the blue curve. However, our results show the opposite, with poisoned samples even displaying greater prediction inconsistency. This indicates that SCALE-UP fails to detect the dormant backdoor in teacher models attacked by our SCAR.

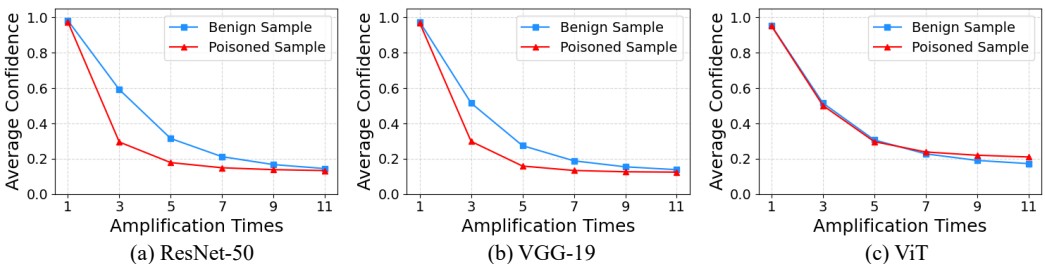

Figure 2: The average confidence of benign and poisoned samples with respect to pixel-wise amplifications on three SCAR-attacked models with different architectures trained on CIFAR-10.

## 5 Conclusion

In this paper, we explored distillation-conditional backdoor attacks (DCBAs), where a dormant backdoor in the teacher model can be activated in the student model through knowledge distillation (KD), even with clean KD data. We found that the direct extension of existing methods (*e.g.*, ADBA (FT)) is ineffective for DCBA and proposed SCAR, which formulates DCBA as a bilevel optimization problem and leverages an implicit differentiation algorithm with a pre-optimized trigger injection function to solve it. Extensive experiments demonstrated the effectiveness of SCAR across diverse datasets, model architectures, and KD techniques, as well as its stealthiness against teacher-side backdoor detection. Our findings reveal a significant yet previously overlooked security threat, and highlight the urgent need to always perform backdoor detection on student models, even when the teacher model and distillation dataset have been verified as secure.

## Acknowledgements

This research is supported in part by the "Pioneer" and "Leading Goose" R&D Program of Zhejiang (2024C01169), the Kunpeng-Ascend Science and Education Innovation Excellence/Incubation Center, the National Natural Science Foundation of China under Grants (62441238, U2441240), and Xiaomi Open-Competition Research Program. This work was mostly done when Yu Yuan and Leyi Qi were Research Assistants at the State Key Laboratory of Blockchain and Data Security, Zhejiang University, China.

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

# Appendix

## A  Why is our SCAR Effective?

In this section, we provide a preliminary analysis of the possible factors contributing to the effectiveness of our SCAR, focusing on two key perspectives: the information gap introduced during knowledge distillation (KD) and the distributional similarity between the training and distillation datasets. Besides, the convergence and approximation error bounds of the implicit differentiation algorithm used in SCAR have been theoretically established in existing studies [25, 65].

### A.1  Analysis from the KD Information Gap

In knowledge distillation (KD), a finite clean dataset is typically used to transfer the teacher model's standard inference behavior to the student model. However, this process may result in an information gap, whereby certain security-relevant characteristics of the teacher model are not fully inherited by the student. This gap can compromise the robustness of the student model.

Recent studies on KD primarily aim to improve the student's performance under capacity constraints by filtering out what is deemed redundant information [15, 41]. While effective for performance, this strategy may inadvertently discard safety-relevant cues that contribute to the teacher model's resilience. In particular, a model's robustness against backdoor attacks often relies not only on task-relevant features but also on seemingly redundant information that plays a defensive role. If the distillation process excludes this information, the resulting student model—though performant—may remain highly susceptible to backdoor threats, even in cases where the teacher model is unaffected.

Therefore, we argue that *the bilevel optimization process of SCAR, which injects distillation-conditional backdoors, may exploit the information gap stemming from incomplete knowledge transfer*. We aim to support this argument through the following proposition.

**Proposition 1.** *Consider knowledge distillation (KD) performed solely on clean samples, denoted by $\mathcal{D}_{clean}$. Let the teacher representation be $T = (Z, \Omega)$ and the student representation be $S = (Z, \Omega_{\parallel})$, where:*

- *$Z$ denotes task-relevant features,*

- *$\Omega = (\Omega_{\parallel}, \Omega_{\perp})$ represents the total backdoor-relevant information,*

- *$\Omega_{\parallel}$ is the component of $\Omega$ that is coupled with $Z$,*

- *$\Omega_{\perp}$ is orthogonal to both $Z$ and $\Omega_{\parallel}$.*

*Then, the following inequality holds:*

$$D_{gap} = I(T; Y_{backdoor}) - I(S; Y_{backdoor}) \geq 0, \tag{14}$$

*where $I(\cdot; \cdot)$ denotes mutual information and $Y_{backdoor}$ is the output label of backdoor samples.*

*This information gap $D_{gap}$ arises because KD on clean data can only transfer the coupled component $\Omega_{\parallel}$, while the orthogonal component $\Omega_{\perp}$—which may carry important safety-relevant signals—remains untransferred.*

Building on the above proposition, we emphasize the critical role of coupling between backdoor-relevant and task-relevant features in the teacher model. This coupling is key to enabling the information gap that SCAR exploits. Specifically, before initiating the bilevel optimization in SCAR, we perform a pre-optimization step to craft the trigger pattern. This step yields a trigger that naturally exists in the clean dataset, making it partially correlated with task-relevant features $Z$.

During training, the teacher model is exposed to both clean and backdoor samples, learning a robust representation that includes $Z$, $\Omega_{\parallel}$, and $\Omega_{\perp}$. We can construct the total backdoor-relevant information $\Omega = (\Omega_{\parallel}, \Omega_{\perp})$ such that $\Omega_{\parallel}$ is coupled with $Z$, while $\Omega_{\perp}$ remains orthogonal. However, the student model is trained through KD using only clean samples. As a result, the student inherits $Z$ and the coupled component $\Omega_{\parallel}$, but not the orthogonal, safety-relevant component $\Omega_{\perp}$ that contributes to the teacher's robustness. This selective transfer induces an information gap $D_{gap} \geq 0$,

enabling a dormant backdoor effect: while the teacher model remains resilient, the student model becomes vulnerable to backdoor attacks. Its proof is as follows.

*Proof.* We begin by applying the chain rule of mutual information to expand the information gap between teacher and student representations.

For the teacher model's representation $T = (Z, \Omega_\parallel, \Omega_\perp)$, we have:

$$\begin{aligned} I(T; Y_{\text{backdoor}}) &= I(Z, \Omega_\parallel, \Omega_\perp; Y_{\text{backdoor}}) \\ &= I(Z; Y_{\text{backdoor}}) + I(\Omega_\parallel, \Omega_\perp; Y_{\text{backdoor}} \mid Z), \end{aligned} \tag{15}$$

where the second equality follows from the chain rule of mutual information.

Similarly, for the student model's representation $S = (Z, \Omega_\parallel)$:

$$\begin{aligned} I(S; Y_{\text{backdoor}}) &= I(Z, \Omega_\parallel; Y_{\text{backdoor}}) \\ &= I(Z; Y_{\text{backdoor}}) + I(\Omega_\parallel; Y_{\text{backdoor}} \mid Z). \end{aligned} \tag{16}$$

Taking the difference between equations (15) and (16) yields:

$$\begin{aligned} D_{\text{gap}} &= I(T; Y_{\text{backdoor}}) - I(S; Y_{\text{backdoor}}) \\ &= I(\Omega_\parallel, \Omega_\perp; Y_{\text{backdoor}} \mid Z) - I(\Omega_\parallel; Y_{\text{backdoor}} \mid Z). \end{aligned} \tag{17}$$

Since the backdoor information can be decomposed as $\Omega = (\Omega_\parallel, \Omega_\perp)$, we apply the chain rule once more to the first term in equation (17):

$$I(\Omega_\parallel, \Omega_\perp; Y_{\text{backdoor}} \mid Z) = I(\Omega_\parallel; Y_{\text{backdoor}} \mid Z) + I(\Omega_\perp; Y_{\text{backdoor}} \mid Z, \Omega_\parallel). \tag{18}$$

Substituting equation (18) back into equation (17):

$$\begin{aligned} D_{\text{gap}} &= \left[ I(\Omega_\parallel; Y_{\text{backdoor}} \mid Z) + I(\Omega_\perp; Y_{\text{backdoor}} \mid Z, \Omega_\parallel) \right] - I(\Omega_\parallel; Y_{\text{backdoor}} \mid Z) \\ &= I(\Omega_\perp; Y_{\text{backdoor}} \mid Z, \Omega_\parallel). \end{aligned} \tag{19}$$

To establish non-negativity, we invoke the fundamental property that conditional mutual information is always non-negative. By definition in [10]:

$$I(A; B \mid C) = H(B \mid C) - H(B \mid C, A). \tag{20}$$

Since conditioning never increases entropy [9], we have:

$$H(B \mid C, A) \leq H(B \mid C), \tag{21}$$

which implies:

$$I(A; B \mid C) \geq 0. \tag{22}$$

Applying this property to equation (19) with $A = \Omega_\perp$, $B = Y_{\text{backdoor}}$, and $C = (Z, \Omega_\parallel)$, we conclude:

$$D_{\text{gap}} = I(\Omega_\perp; Y_{\text{backdoor}} \mid Z, \Omega_\parallel) \geq 0. \tag{23}$$

$\square$

From the perspective of the information gap, we intended to suggest that *SCAR may exploit the inherent discrepancies arising from imperfect knowledge transfer during knowledge distillation (KD) to activate backdoors in the student model*. We note that, perhaps counterintuitively, such information gaps are intrinsic to the KD process, as the student model is unlikely to precisely replicate the behavior of the teacher model. This divergence may stem from architectural differences, limited model capacity, or the inherently lossy nature of the distillation objective [8, 24]. As such, while the teacher model may behave benignly, the student model can still manifest backdoor behaviors.

### A.2 Analysis from the KD Data Distribution

Beyond the information gap perspective, we also provide an empirical analysis from the standpoint of the data distribution used during KD. We hypothesize that *using a benign dataset drawn from the same distribution as the teacher model's training data may cause the student model to implicitly learn data-distribution-specific natural backdoor features, thereby facilitating backdoor injection.* Specifically, during the pre-optimization of the trigger injection function, our goal is to obtain a natural trigger pattern (NTP) that can persist throughout the KD process. This NTP resembles a targeted universal adversarial perturbation (TUAP) that consistently perturbs samples from the given data distribution toward an attacker-specified target label. Existing research on adversarial attacks suggests that the transferability of adversarial perturbations is partially influenced by model architecture and output distribution similarity [89]. We also observe several experimental phenomena that may serve as supporting evidence for the above hypothesis:

**(1)** In our CIFAR-10 experiments, the teacher models typically achieve over 90% accuracy on the test set, which results in output probabilities that closely align with the one-hot ground-truth labels. Therefore, when using such high-accuracy teacher models for KD, the student model is likely to inherit knowledge related to the data distribution itself—similar to what a model would learn when trained from scratch. Since NTPs (like TUAPs) exhibit strong transferability between models with highly similar output distributions, any model that learns data-distribution-specific knowledge and achieves high accuracy is likely to be vulnerable to such NTP attacks.

**(2)** A possible explanation for the degraded performance of SCAR on the ImageNet dataset is that: the relatively low accuracy (typically $< 70\%$) of models on ImageNet leads to less similar output distributions across models. This probably prevents the student model from accurately capturing data-distribution-specific knowledge, and reduces the transferability and effectiveness of the NTP.

**(3)** Meanwhile, we observe that SCAR is capable of successfully attacking student models across various distillation methods, not limited to the KL-divergence-based distillation used during the bilevel optimization process. This suggests that the transferability of the NTP between models with the same data-distribution-specific knowledge may play a significant role in facilitating the attack.

From the data distribution perspective, we intended to suggest that *SCAR may leverage distribution-specific natural backdoor features to facilitate the persistence of backdoors across different KD processes*. Regardless of the KD method employed, the primary objective is to optimize the student model's performance on the given data distribution. However, during this process, the student model distilled from datasets drawn from the same distribution as the teacher may also implicitly learn certain distribution-specific natural backdoor features. This might allow the backdoor trigger to persist across different KD strategies.

## B Results on Additional Model Architectures

In this section, we conduct experiments on two additional teacher model architectures, including VGG-19 [71] and ViT [19]. We conduct experiments on the CIFAR-10 dataset. The KD methods, attack setup, and baseline methods are the same as those described in the main text.

As shown in Table 4 and 5, SCAR demonstrates superior attack performance compared to the baseline method ADBA (FT). Specifically, on the VGG-19 teacher model, ADBA (FT) exhibits only limited backdoor injection performance against the student model ShuffleNet-V2, and even fails entirely to implant a backdoor into EfficientViT. In contrast, SCAR consistently achieves a higher attack success rate above 80% against student models in most scenarios. These results further validate the effectiveness of our proposed SCAR method.

## C Results on Additional Backdoor Detection Methods

In this section, we further verify the stealthiness of the distillation-conditional backdoor injected by SCAR in the teacher model by employing various advanced backdoor detection methods.

Table 4: The attack performance (%) of SCAR compared to baseline methods on the VGG-19 teacher model and its three distilled student models. For each dataset under each KD method, the best results are **boldfaced**, while all failed cases (student ASR < 50%) are marked in red.

| KD Method | Model | VGG-19 (Teacher) | | MobileNet-V2 (Student A) | | ShuffleNet-V2 (Student B) | | EfficientViT (Student C) | |
|---|---|---|---|---|---|---|---|---|---|
| | Attack | ACC | ASR↓ | ACC | ASR↑ | ACC | ASR↑ | ACC | ASR↑ |
| Response | Benign | 92.41 | 0 | 91.07 | 0 | 90.24 | 0 | 87.56 | 0 |
| | ADBA (FT) | 90.58 | 6.88 | 91.37 | 92.39 | 86.19 | 61.12 | 87.26 | 47.39 |
| | SCAR | 90.53 | 1.50 | 91.43 | **99.24** | 89.17 | **95.14** | 86.26 | **90.36** |
| Feature | Benign | 92.41 | 0 | 91.54 | 0 | 90.60 | 0 | 87.76 | 0 |
| | ADBA (FT) | 90.58 | 6.88 | 91.41 | 90.03 | 89.50 | 75.60 | 85.33 | 32.22 |
| | SCAR | 90.53 | 1.50 | 91.54 | **99.66** | 89.05 | **96.20** | 85.55 | **72.64** |
| Relation | Benign | 92.41 | 0 | 91.81 | 0 | 89.43 | 0 | 87.36 | 0 |
| | ADBA (FT) | 90.58 | 6.88 | 91.01 | 91.11 | 85.76 | 46.14 | 86.99 | 44.77 |
| | SCAR | 90.53 | 1.50 | 91.27 | **99.72** | 89.39 | **97.23** | 86.12 | **88.47** |

Table 5: The attack performance (%) of SCAR compared to baseline methods on the ViT teacher model and its three distilled student models. For each dataset under each KD method, the best results are **boldfaced**, while all failed cases (student ASR < 50%) are marked in red.

| KD Method | Model | ViT (Teacher) | | MobileNet-V2 (Student A) | | ShuffleNet-V2 (Student B) | | EfficientViT (Student C) | |
|---|---|---|---|---|---|---|---|---|---|
| | Attack | ACC | ASR↓ | ACC | ASR↑ | ACC | ASR↑ | ACC | ASR↑ |
| Response | Benign | 84.92 | 0 | 85.10 | 0 | 84.16 | 0 | 84.52 | 0 |
| | ADBA (FT) | 83.74 | 4.33 | 82.75 | 83.56 | 82.27 | 80.61 | 83.12 | 51.83 |
| | SCAR | 84.71 | 1.73 | 85.35 | **95.01** | 84.06 | **93.90** | 84.10 | **84.58** |
| Feature | Benign | 84.92 | 0 | 84.41 | 0 | 85.31 | 0 | 84.23 | 0 |
| | ADBA (FT) | 83.74 | 4.33 | 83.75 | 86.64 | 83.63 | 87.53 | 83.06 | 69.91 |
| | SCAR | 84.71 | 1.73 | 85.64 | **89.54** | 85.03 | **89.74** | 84.23 | **88.19** |
| Relation | Benign | 84.92 | 0 | 85.18 | 0 | 84.97 | 0 | 84.55 | 0 |
| | ADBA (FT) | 83.74 | 4.33 | 83.34 | 85.87 | 83.33 | 71.44 | 83.76 | 48.48 |
| | SCAR | 84.71 | 1.73 | 85.94 | **98.27** | 85.86 | **87.10** | 85.23 | **80.87** |

## C.1 The Resistance to BTI-DBF

BTI-DBF [82] is an advanced backdoor trigger inversion (BTI)-based backdoor detection method. Overall, its detection process consists of three main steps. First, a mask is trained at the feature level to decouple the benign features involved in the inference of clean samples from the remaining redundant features. Next, using this mask, a backdoor generator is trained to minimize the discrepancy of benign features between clean samples and their corresponding poisoned versions, while maximizing the discrepancy of their backdoor features. Finally, the trained backdoor generator is applied to the test set to observe the distribution of the model's output classes. If the model exhibits a strong bias toward a specific label, that label is identified as a potential backdoor target.

We train teacher models with three different architectures on the CIFAR-10 dataset utilizing the SCAR attack, with the target label set to class 0. These models are then evaluated utilizing the BTI-DBF detection method, and the distribution of predicted class labels on the test set is recorded for each model. As shown in Table 6, BTI-DBF fails to effectively detect the distillation-conditioned backdoor embedded in the teacher models by SCAR. Specifically, for the VGG-19 and ResNet-50 models, BTI-DBF mistakenly identifies Class 2 as the target label, while for the ViT model, it fails to detect any backdoor at all. These experimental results demonstrate that SCAR can effectively evade detection by BTI-DBF.

Table 6: The backdoor detection performance of BTI-DBF on teacher models compromised by SCAR. Class 0 is the attack-defined target label. False detections are marked in red.

| Model | Predicted Number of Each Class (> 5000 indicates a potential backdoor) | | | | | | | | | |
|---|---|---|---|---|---|---|---|---|---|---|
| | **Class 0** | Class 1 | Class 2 | Class 3 | Class 4 | Class 5 | Class 6 | Class 7 | Class 8 | Class 9 |
| ResNet-50 | 140 | 116 | 7113 | 1 | 1 | 0 | 0 | 6 | 1616 | 1007 |
| VGG-19 | 0 | 0 | 5610 | 0 | 72 | 4318 | 0 | 0 | 0 | 0 |
| ViT | 854 | 1013 | 928 | 900 | 1069 | 998 | 1093 | 1038 | 977 | 1130 |

Table 7: The SAP values reported by A2D on teacher models compromised by SCAR.

| Trial | 1 | 2 | 3 | 4 | 5 |
|---|---|---|---|---|---|
| SAP | 0.117 | 0.117 | 0.185 | 0.206 | 0.027 |

Table 8: The backdoor detection performance of BAN on teacher models compromised by SCAR. Class 0 is the attack-defined target label. False detections are marked in red.

| Trial | Predicted Number of Each Class (maximum indicates the target label inferred by BAN) | | | | | | | | | |
|---|---|---|---|---|---|---|---|---|---|---|
| | **Class 0** | Class 1 | Class 2 | Class 3 | Class 4 | Class 5 | Class 6 | Class 7 | Class 8 | Class 9 |
| 1 | 434 | 461 | 1943 | 459 | 43 | 558 | 151 | 46 | 454 | 451 |
| 2 | 333 | 2628 | 272 | 70 | 146 | 38 | 0 | 253 | 766 | 494 |
| 3 | 141 | 432 | 260 | 1028 | 11 | 6 | 111 | 2057 | 536 | 418 |
| 4 | 297 | 43 | 215 | 266 | 3066 | 287 | 228 | 89 | 285 | 224 |
| 5 | 176 | 371 | 696 | 79 | 65 | 1182 | 123 | 1512 | 209 | 587 |

## C.2 The Resistance to A2D

A2D [20] is a black-box backdoor detection framework that exploits the heightened sensitivity of backdoored models to adversarial perturbations. It introduces the sensitivity to adversarial perturbations (SAP) metric, estimated by transferring strong adversarial attacks generated on an unrelated reference model. A model is flagged as trojaned if its SAP exceeds a threshold, requiring only limited clean data and low computational cost.

To further verify the stealthiness of our SCAR, we conduct 5 independent detection trials on a ResNet-50 teacher trained on CIFAR-10 with target label 0. As shown in Table 7, the SCAR-attacked teacher consistently exhibits low SAP, indicating that A2D failed to detect the backdoor.

## C.3 The Resistance to BAN

BAN [81] enhances backdoor detection by augmenting backdoor feature inversion with neuron activation information. By adversarially perturbing model weights to trigger backdoor effects, BAN enables efficient and accurate separation of benign and backdoored models.

To further verify the stealthiness of our SCAR, we conduct 5 independent detection trials on a ResNet-50 teacher trained on CIFAR-10 with target label 0. Similar to BTI-DBF, we record both the prediction distribution after mask training and the predicted target class. As shown in Table 8, BAN also consistently misjudges the backdoor target class, leading to a high false positive rate.

## C.4 The Resistance to MDTD and TED

In this section, we verify the stealthiness of our SCAR utilizing two advanced input-level detection methods, MDTD [67] and TED [58]. Specifically, MDTD is a universal test-time backdoor detector for DNNs that requires no prior knowledge of the trigger strategy. It leverages adversarial learning to estimate distances to the decision boundary, identifying poisoned inputs as those lying

Table 9: F1-scores of MDTD and TED on poisoned test samples for the SCAR-attacked ResNet-50 teacher model.

| Method | 1 | 2 | 3 | 4 | 5 |
|--------|--------|--------|--------|--------|--------|
| MDTD | 0.0488 | 0.0417 | 0.0225 | 0.0512 | 0.0580 |
| TED | 0.0386 | 0.0281 | 0.0056 | 0.0120 | 0.0196 |

farther away than clean samples. TED is a model-agnostic backdoor detector that treats DNNs as dynamical systems. Instead of relying on separability in a metric space, TED traces the evolution of inputs: benign samples follow consistent trajectories, whereas poisoned ones diverge towards attacker-specified targets, revealing the presence of backdoors.

We evaluate MDTD and TED on a ResNet-50 teacher trained on CIFAR-10 (target label 0) over 5 independent trials. Since both methods are input-level detection, we utilize the poisoned test set as input and compute the F1-score for poisoned sample detection. As shown in Table 9, both methods yield very low F1-scores ($< 0.06$), indicating that they misclassify most poisoned samples as benign and thus failed to detect the poisoning effectively.

## D    Results on Additional Non-Detection Backdoor Defenses

In our threat model, the attacker uploads a model containing a distillation-conditional backdoor to a trusted third-party platform. This platform performs backdoor detection and, upon verifying no abnormal behavior, releases the model. A victim user then downloads the model and uses it for further KD. A key aspect of this scenario is that the platform is usually only responsible for detection, not mitigation. Typically, uploaded models may be trained on large-scale datasets over long periods, while developers usually only upload their model without uploading training data and configurations. As such, such mitigation is generally infeasible or at least significantly degrades model performance and requires many computational resources. Given the platform's limited resources, we assume it does not conduct non-detection defenses.

However, users may apply backdoor mitigation methods either to the downloaded teacher model before KD or to the distilled student model afterward. We conduct supplementary experiments to evaluate the robustness of SCAR in both scenarios.

### D.1    The Resistance to Teacher-side Mitigations

In this section, we conduct additional non-detection backdoor defenses to SCAR-attacked teacher models to verify the robustness of our SCAR. We evaluate two non-detection defenses: NAD [45] and fine-pruning [49]. We apply these defenses to a SCAR-attacked ResNet-50 on CIFAR-10, and then use the defended model for KD to train MobileNet-V2. As shown in Table 10, SCAR remains highly effective, even after the teacher has undergone these backdoor mitigations. Besides, using non-detection defenses may even slightly raise teacher ASR ($> 5\%$). This might be because such defenses disrupt the carefully crafted "mask" of the distillation-conditional backdoor, thus re-exposing the hidden backdoor to some extent.

While NAD and fine-pruning fail to eliminate SCAR, we don't intend to claim that it is undefeatable. Rather, our goal is to raise awareness that even models which pass backdoor detection should always not be assumed safe for distillation without further examination or cleansing.

### D.2    The Resistance to Student-side Mitigations

In this section, we conduct additional non-detection backdoor defenses to student models distilled from SCAR-attacked teacher models to verify the robustness of our SCAR. We evaluate our SCAR under Sampdetox [84] method. Specifically, we launch the SCAR attack on a ResNet-50 teacher model on CIFAR-10, and distill three MobileNet-V2 student models using different KD methods. We then evaluate the defense performance of SampDetox using both benign and poisoned test sets.

As shown in Table 11, although SampDetox can partially reduce the attack success rate of SCAR, it also leads to a noticeable drop in benign accuracy (BA). This suggests that SCAR has already

Table 10: The attack performance (%) of SCAR on the student MobileNet-V2 distilled from the teacher ResNet-50 under two backdoor mitigation methods, using three KD methods.

| Method | ResNet-50 (w/o Defense) | | ResNet-50 (w/ Defense) | | MobileNet-V2 (Response) | | MobileNet-V2 (Feature) | | MobileNet-V2 (Relation) | |
|---|---|---|---|---|---|---|---|---|---|---|
| | ACC | ASR↓ | ACC | ASR↓ | ACC | ASR↑ | ACC | ASR↑ | ACC | ASR↑ |
| No Defense | 92.47 | 1.50 | — | — | 91.62 | 99.94 | 91.01 | 99.90 | 91.29 | 99.93 |
| NAD | 92.47 | 1.50 | 88.45 | 9.60 | 89.08 | 99.94 | 86.12 | 89.20 | 87.11 | 99.58 |
| fine-pruning | 92.47 | 1.50 | 92.72 | 5.94 | 92.17 | 99.47 | 90.44 | 97.41 | 91.08 | 99.34 |

Table 11: The defense performance (%) of SampDetox against MobileNet-V2 student models distilled from ResNet-50 teacher models using three different KD methods.

| Distillation Method | Before Defense | | After Defense | |
|---|---|---|---|---|
| | ACC | ASR | ACC | ASR |
| Response | 91.62 | 99.94 | 84.13 | 56.42 |
| Feature | 91.01 | 99.90 | 82.26 | 52.60 |
| Relation | 91.29 | 99.93 | 83.65 | 76.01 |

Table 12: The attack performance (%) of SCAR with different surrogate models on the student MobileNet-V2 distilled from the teacher ResNet-50 using three KD methods.

| Surrogate | ResNet-50 (Teacher) | | MobileNet-V2 (Response) | | MobileNet-V2 (Feature) | | MobileNet-V2 (Relation) | |
|---|---|---|---|---|---|---|---|---|
| | ACC | ASR↓ | ACC | ASR↑ | ACC | ASR↑ | ACC | ASR↑ |
| ResNet-18 | 92.47 | 1.50 | 91.62 | 99.94 | 91.01 | 99.90 | 91.29 | 99.93 |
| ResNet-34 | 92.32 | 1.42 | 92.26 | 99.47 | 90.77 | 99.68 | 91.76 | 99.92 |
| DenseNet-121 | 92.79 | 1.43 | 92.19 | 99.80 | 90.67 | 95.78 | 91.80 | 99.90 |
| SqueezeNet | 92.41 | 1.56 | 91.86 | 99.94 | 91.16 | 99.13 | 91.78 | 99.88 |

exhibited a certain degree of robustness against input pre-processing defenses, although we have no particular design for it.

# E  Results on Additional Ablation Studies

## E.1  The Surrogate Model Architecture

In this section, we evaluate SCAR on CIFAR-10 under various surrogate architectures, including ResNet-18, ResNet-34, DenseNet-121, and SqueezeNet, while adopting ResNet-50 and MobileNet-V2 as the teacher and student models, respectively. As shown in Table 12, SCAR consistently achieves high attack success rates (student ASR > 95%) across all settings.

## E.2  The Distillation Loss Weight $\delta$

In our threat model, the attacker can only manipulate the training process of the compromised teacher model, without access to the user's choice of distillation methods or hyperparameters. In this section, we evaluate the effectiveness of SCAR in attacking student models under various distillation methods and loss weights $\delta$. Specifically, we utilize a SCAR-compromised ResNet-50 trained on CIFAR-10 as the teacher model and distill it into MobileNet-V2 student models using three different distillation techniques and five different distillation loss weights $\delta$.

Table 13: The attack performance (%) of SCAR under different distillation loss weights ($\delta$) utilizing the student model (MobileNet-V2) distilled from the SCAR-attacked teacher model (ResNet-50).

| Model | ResNet-50 (Teacher) | | MobileNet-V2 ($\delta = 1$) | | MobileNet-V2 ($\delta = 2$) | | MobileNet-V2 ($\delta = 3$) | | MobileNet-V2 ($\delta = 4$) | | MobileNet-V2 ($\delta = 5$) | |
|---|---|---|---|---|---|---|---|---|---|---|---|---|
| KD Method | ACC | ASR$\downarrow$ | ACC | ASR$\uparrow$ | ACC | ASR$\uparrow$ | ACC | ASR$\uparrow$ | ACC | ASR$\uparrow$ | ACC | ASR$\uparrow$ |
| Response | 92.47 | 1.50 | 91.62 | 99.94 | 91.64 | 99.92 | 91.58 | 99.87 | 91.68 | 99.90 | 92.15 | 99.96 |
| Feature | 92.47 | 1.50 | 91.01 | 99.90 | 90.68 | 98.70 | 91.27 | 98.17 | 91.12 | 94.00 | 91.11 | 95.81 |
| Relation | 92.47 | 1.50 | 91.29 | 99.93 | 91.65 | 99.79 | 91.27 | 99.89 | 91.47 | 99.91 | 91.70 | 99.96 |

Table 14: The attack performance (%) of SCAR on the student MobileNet-V2 distilled from the teacher ResNet-50 with different distillation datasets, using three KD methods.

| Distillation Dataset | ResNet-50 (Teacher) | | MobileNet-V2 (Response) | | MobileNet-V2 (Feature) | | MobileNet-V2 (Relation) | |
|---|---|---|---|---|---|---|---|---|
| | ACC | ASR$\downarrow$ | ACC | ASR$\uparrow$ | ACC | ASR$\uparrow$ | ACC | ASR$\uparrow$ |
| CIFAR-10 | 92.47 | 1.50 | 91.62 | 99.94 | 91.01 | 99.90 | 91.29 | 99.93 |
| CINIC-10 | 68.83 | 5.22 | 76.22 | 74.51 | 72.35 | 54.83 | 75.81 | 75.58 |

As shown in Table 13, SCAR consistently achieves effective attacks on the MobileNet-V2 student models across different distillation methods and loss weights $\delta$. This further demonstrates the ability of SCAR to activate the dormant backdoor in the teacher model and achieve a high attack success rate in the student model.

### E.3 The Distillation Dataset

In addition to the distillation loss weight $\delta$, users can also specify the dataset employed for distilling the student model. Here, we conduct additional experiments where the distillation dataset differs from the training set. We train a ResNet-50 teacher using SCAR on CIFAR-10, and then distill MobileNet-V2 students using a subset of CINIC-10 [12], which contains the same 10 classes but has a different data distribution (5,000 training and 1,000 test samples per class).

As shown in Table 14, due to the distributional shift, the teacher's accuracy drops to 68.83%. The students also experience a drop in accuracy. Although the attack success rate on the students decreased accordingly, it still remained above 54%. These results suggest that if the student fails to achieve high accuracy during distillation—meaning it does not effectively inherit the teacher's benign knowledge—it is also less likely to inherit the backdoor. However, such significant drops in accuracy for both teacher and student models indicates an ineffective distillation process, which is usually not meaningful in practice.

### E.4 The Number of Inner Steps $T$

In our main experiments, We fix the number of inner optimization steps to 20. This is mostly because the surrogate model has already reached a reasonably good local optimum (around 85% accuracy compared to the teacher) within these steps, which is sufficient for guiding the outer optimization.

To assess sensitivity to this setting, we conduct ablation studies using different inner steps (20, 30, 40) when attacking ResNet-50 on CIFAR-10. The surrogate model is ResNet-18, and the student model is MobileNet-V2. As shown in Table 15, SCAR remains effective under all settings. This suggests that once the inner parameters are optimized to a nearly local optimum, the outer gradient is adequately approximated.

### E.5 The Bilevel Optimization Hyperparameters ($\alpha$, $\beta$, and $\gamma$)

In the outer optimization objective in Eq. (2), the scales of the four loss terms are approximately comparable and relate to four different aspects (*e.g.*, the student's performance on benign samples).

Table 15: The attack performance (%) of SCAR with different inner steps on the student MobileNet-V2 distilled from the teacher ResNet-50 using three KD methods.

| Inner Step | ResNet-50 (Teacher) | | MobileNet-V2 (Response) | | MobileNet-V2 (Feature) | | MobileNet-V2 (Relation) | |
|---|---|---|---|---|---|---|---|---|
| | ACC | ASR↓ | ACC | ASR↑ | ACC | ASR↑ | ACC | ASR↑ |
| 20 steps | 92.47 | 1.50 | 91.62 | 99.94 | 91.01 | 99.90 | 91.29 | 99.93 |
| 30 steps | 92.35 | 1.46 | 92.13 | 99.41 | 91.03 | 99.04 | 91.26 | 99.39 |
| 40 steps | 92.20 | 1.51 | 91.98 | 99.69 | 91.16 | 99.18 | 91.87 | 99.49 |

Table 16: The attack performance (%) of SCAR on the student MobileNet-V2 distilled from the teacher ResNet-50 under different hyperparameter settings, using three KD methods.

| Hyperparameter | ResNet-50 (Teacher) | | MobileNet-V2 (Response) | | MobileNet-V2 (Feature) | | MobileNet-V2 (Relation) | |
|---|---|---|---|---|---|---|---|---|
| | ACC | ASR↓ | ACC | ASR↑ | ACC | ASR↑ | ACC | ASR↑ |
| $\alpha = 0.5$ | 92.43 | 1.77 | 91.67 | 99.81 | 91.12 | 99.36 | 91.58 | 99.94 |
| $\alpha = 1$ | 92.47 | 1.50 | 91.62 | 99.94 | 91.01 | 99.90 | 91.29 | 99.93 |
| $\alpha = 1.5$ | 92.54 | 1.33 | 92.08 | 99.46 | 91.03 | 99.71 | 91.78 | 99.59 |
| $\beta = 0.5$ | 92.21 | 1.61 | 92.36 | 99.37 | 91.30 | 99.04 | 91.33 | 99.07 |
| $\beta = 1$ | 92.47 | 1.50 | 91.62 | 99.94 | 91.01 | 99.90 | 91.29 | 99.93 |
| $\beta = 1.5$ | 92.35 | 1.43 | 91.99 | 99.83 | 90.87 | 99.01 | 91.52 | 99.36 |
| $\gamma = 0.5$ | 92.55 | 1.82 | 92.40 | 99.51 | 91.25 | 98.58 | 91.46 | 99.53 |
| $\gamma = 1$ | 92.47 | 1.50 | 91.62 | 99.94 | 91.01 | 99.90 | 91.29 | 99.93 |
| $\gamma = 1.5$ | 92.13 | 1.88 | 91.74 | 99.97 | 90.98 | 99.77 | 91.64 | 99.23 |

Attackers can adjust their values based on specific needs or priorities. In our experiments, we set $\alpha = \beta = \gamma = 1$ for simplicity and clarity, as this is the most straightforward configuration.

We also conduct ablation studies to evaluate SCAR's sensitivity to these hyperparameters. Specifically, we train teachers (ResNet-50) using SCAR on CIFAR-10 with $\alpha$, $\beta$, and $\gamma$ independently set to 0.5, 1.0, and 1.5. We use ResNet-18 as the surrogate and MobileNet-V2 as the student. As shown in Table 16, SCAR consistently achieves strong attack performance (student ASR $> 98\%$) across different hyperparameter settings.

### E.6 The Trigger Injection Function $G$

In this section, we consider more types of triggers to replace the pre-optimized trigger function $G$, especially the sample-specified triggers cover the entire image. We replace the pre-trained triggers in SCAR with image-size poisoning strategies from BadNets [27], WaNet [60] and BppAttack [76], as implemented in `BackdoorBox` [47] or their source codes. We then train teacher models ResNet-50 on CIFAR-10 using these methods. ResNet-18 is used as the surrogate, and MobileNet-V2 serves as the student. We evaluate the attack success rates on students obtained via different KD methods.

As shown in Table 17, using these image-size poisoning strategies still fails to effectively transfer backdoors to the student models (student ASR $< 25\%$). It indicates that simply using image-size triggers cannot be easily 'entangled' with benign features, even if they cover the entire image, and thus fail to transfer during distillation.

## F Implementation Details

**Details of Datasets.**

1. *CIFAR-10.* The CIFAR-10 dataset [40] contains 50,000 training samples and 10,000 testing samples in total. The dataset has 10 classes and each class has 5,000 training samples and 1,000 testing samples. The size of each image sample is $3 \times 32 \times 32$.

Table 17: The attack performance (%) of SCAR on the student MobileNet-V2 distilled from the teacher ResNet-50 under different trigger types, using three KD methods.

| Trigger Type | ResNet-50 (Teacher) | | MobileNet-V2 (Response) | | MobileNet-V2 (Feature) | | MobileNet-V2 (Relation) | |
|---|---|---|---|---|---|---|---|---|
| | ACC | ASR↓ | ACC | ASR↑ | ACC | ASR↑ | ACC | ASR↑ |
| BadNets | 93.81 | 0.72 | 91.47 | 1.03 | 91.49 | 1.48 | 91.22 | 1.38 |
| WaNet | 93.07 | 2.20 | 91.07 | 20.97 | 90.47 | 9.73 | 90.77 | 21.22 |
| BppAttack | 93.65 | 1.11 | 91.43 | 1.17 | 89.29 | 1.86 | 91.36 | 1.07 |
| **Ours** | 92.47 | 1.50 | 91.62 | 99.94 | 91.01 | 99.90 | 91.29 | 99.93 |

2. *ImageNet.* The ImageNet dataset [14] consists of 1,000 classes containing over 14 million manually annotated images. In this paper, we select a subset with 50 different classes and each class contains 500 training samples and 100 testing samples with size $3 \times 224 \times 224$.

**Details of Training Models.**

1. *Benign Models.* We utilize the Adam as the optimizer and the batch size is set to 256 on CIFAR-10 and 128 on ImageNet. We set the initial learning rate as $10^{-4}$ and train all models for 200 epochs, with the learning rate reduced by a cosine annealing schedule.

2. *Models attacked by ADBA (FT).* We first train a teacher model compromised by ADBA using the same training configuration as that of the benign model, with the target label set to 0. The shadow model is trained via KL-divergence-based distillation using SGD with an initial learning rate of 0.1. The trigger pattern is optimized using a separate SGD optimizer with an initial learning rate of 0.01. All optimization processes adopt individual cosine annealing schedules to manage the learning rates. During fine-tuning, the feature extraction layers of the teacher model are frozen, and only the final linear layer is updated. The optimizer used is Adam with a learning rate of $10^{-6}$, and the hyperparameter $\eta$ is set to 1 and $k$ is set to 0.1.

3. *Models attacked by SCAR.* During trigger pre-optimization, we distill a benign model using KL divergence loss with the Adam optimizer (learning rate of $10^{-4}$) and a cosine annealing schedule. The trigger pattern is then optimized using Adam with a learning rate of 0.01. For solving the bilevel optimization problem, we set the target label to 0. The outer optimization runs for 200 iterations, with 20 inner-loop updates per outer iteration, and each fixed-point iteration runs for 100 steps. The teacher model is optimized using Adam with an initial learning rate of $10^{-4}$, and the learning rate is decayed using a cosine annealing schedule. To accelerate training, 40 random batches are selected from the training set in each fixed-point iteration. The hyperparameters, $\alpha$, $\beta$, $\gamma$, and $\delta$, are set to 1.

**Details of Knowledge Distillation.** When distilling student models to evaluate the effectiveness of SCAR, we set the hyperparameter $\delta$ to 1 for all three distillation methods. Adam is used as the optimizer with an initial learning rate of $10^{-4}$, and the learning rate is scheduled using cosine annealing over 150 epochs. We apply early stopping once the student model achieves comparable or slightly better accuracy than the teacher model on the test set.

**Computational Resources.** In our implementations, we utilize PyTorch as the deep learning framework. All our experiments are implemented with RTX 4090 GPUs.

# G   The Overhead of our SCAR

In this section, we evaluate the overhead of our SCAR. Specifically, we separately measure the time of the bilevel optimization process for compromised teacher models on the CIFAR-10 and ImageNet datasets. We set the inner loop to 20 iterations and the outer loop to 200 iterations, with 100 fixed-point updates per outer iteration. To accelerate training, 40 random batches are selected from the training set in each fixed-point iteration. We conduct all training using a single RTX 4090 GPU. For the subsequent KD process, since the attacker can only manipulate the training of the teacher model but cannot control which distillation method the user adopts, we do not account for its overhead.

Table 18: The overhead (hours) of ResNet-50 compromised by SCAR on CIFAR-10 and ImageNet.

| Dataset | CIFAR-10 | ImageNet |
|---------|----------|----------|
| Overhead | 20.75 | 59.04 |

As shown in Table 18, the optimization of the teacher model incurs substantial overhead. This is largely because each fixed-point iteration to estimate its parameter gradients necessitates a 100-iteration linear loop, and furthermore, every iteration within this loop requires gradient computations for large tensors whose dimensions are comparable to the full model parameters. To accelerate training, an intuitive approach is to utilize multi-GPU training, which allows for increasing the batch size in each optimization round. This, in turn, can reduce the number of outer optimization epochs and fixed-point iterations, thereby speeding up the convergence of the teacher model optimization. How to address this training overhead presents a potential direction for our future work.

## H  Related Work

### H.1  Backdoor Attack

Based on whether activating the backdoor requires preconditions beyond the trigger pattern, attacks are classified into two types: **(1)** *Unconditional backdoor attacks* [48, 66, 79], the conventional form, can activate malicious behaviors solely by applying predefined trigger patterns. **(2)** *Conditional backdoor attacks* [38, 62, 72], a more recent paradigm, require specific model manipulations before the trigger pattern can activate the hidden malicious functionality.

**Unconditional Backdoor Attacks.** This class of attacks is the conventional form of backdoors: a predefined trigger pattern is simply added to inputs to induce the model to execute an attacker-specified behavior (e.g., map to a target label) without any additional conditions. The BadNets attack [27], which uses a white-square trigger, is the first to demonstrate the threat of backdoors in the image domain. Subsequent work has produced varied trigger modalities, including pattern blending [5] and imperceptible perturbations [60]. More recently, Cai et al. [2] propose using pitch and timbre as triggers in the audio domain. These unconditional backdoor attacks are simple and highly effective, but they are also relatively easy to detect.

**Conditional Backdoor Attacks.** Generally, models compromised by conditional backdoor attacks initially behave normally, even on triggered inputs, with the backdoor remaining inactive. This latency can be broken by specific post-processing operations applied to the model. Tian et al. [72] first demonstrate that model compression techniques, specifically model pruning and quantization, can awaken hidden backdoors. Subsequently, further research [31, 38, 53, 63] highlights the vulnerability of model quantization to backdoors and proposes various quantization-conditional backdoor attacks. Furthermore, fine-tuning on downstream tasks [39, 62] and dynamic multi-exit transformation [18] have also been shown to activate such backdoors. However, existing studies have not explored whether the KD process might introduce similar security vulnerabilities.

### H.2  Backdoor Defenses

To mitigate backdoor threats, researchers have proposed various defenses, broadly categorized as: **(1)** *Detection-based defenses* [74, 80, 82], aiming to identify whether a model contains a backdoor. **(2)** *Purification-based defenses* [6, 33, 88], seeking to cleanse or inactive potential backdoors within the model. While effective against conventional backdoor attacks, these defenses face challenges with conditional backdoors. Specifically, the dormant nature of conditional backdoors often allows them to evade detection-based methods [53, 72], leading to compromised models being misclassified as benign. Although purification-based approaches might partially reduce the attack success rate (ASR) of conditional backdoors [44], they typically incur additional overhead from necessary model parameter modifications or auxiliary module training. In particular, specialized defenses have recently been developed for conditional backdoors, such as those targeting quantization-conditional backdoors [43, 44], and have demonstrated promising effectiveness. However, defenses against our proposed distillation-conditional backdoor remain largely underexplored.

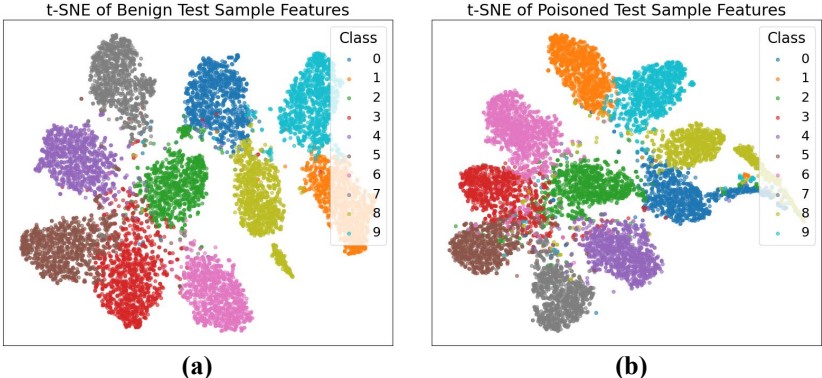

Figure 3: The t-SNE of benign and poisoned test sample features on the teacher model ResNet-50 compromised by SCAR.

# I  Visualization

**The t-SNE of Features on the Teacher Model.** To further verify the stealthiness of SCAR-injected backdoors in the teacher model, we visualize the feature distributions of both benign and poisoned test samples. Specifically, we utilize a teacher model ResNet-50 trained on the CIFAR-10 dataset under SCAR attack. During testing, we feed the model with both benign and poisoned samples, extract their features prior to the fully connected (fc) layer, and apply t-SNE to project them into a two-dimensional space for visualization.

As shown in Figure 3, the ResNet-50 model attacked by SCAR produces well-separated clusters in the feature space for both benign and poisoned test samples. This demonstrates the stealthiness and non-activatability of the SCAR-injected backdoors in the teacher model, enabling the distillation-conditional backdoor to effectively evade direct backdoor detection methods applied to the teacher.

**The t-SNE of Features on the Student Model.** To further validate the effectiveness of SCAR in attacking student models, we visualize the feature distributions of both benign and poisoned test samples as processed by the student model. Specifically, we consider a scenario in which a ResNet-50 teacher model, trained on CIFAR-10 and compromised by SCAR, is used to distill knowledge to a MobileNet-V2 student model via response-based distillation on clean data. During testing, we feed the student model MobileNet-V2 with both benign and poisoned samples, extract their features prior to the fc layer, and apply t-SNE to project them into a two-dimensional space for visualization.

As shown in Figure 4, the benign samples form well-separated clusters in the feature space of the MobileNet-V2 student model, whereas the majority of poisoned samples are misclassified into the target label (Class 0), demonstrating the effectiveness of SCAR in attacking student models.

**The Examples of Benign and Poisoned Images.** In addition, as shown in Figure 5, we present a set of benign images from the CIFAR-10 test set along with their corresponding poisoned versions.

# J  Societal Impact

This paper uncovers a significant and previously underappreciated security vulnerability: student models can inherit backdoor threats from teacher models via knowledge distillation, even if the teacher model has passed prior backdoor detection and distillation is performed using a clean dataset. Our proposed attack, SCAR, though straightforward, effectively demonstrates the practical feasibility of this threat vector. While the techniques presented could potentially be exploited by malicious actors to craft teacher models embedded with distillation-conditional backdoors, the primary contribution of this work is to serve as a crucial alert for the AI development and deployment community. It highlights a critical blind spot in current security practices.

We strongly advocate that developers and organizations implement rigorous security verification, including comprehensive backdoor detection, not only for teacher models sourced from third-party

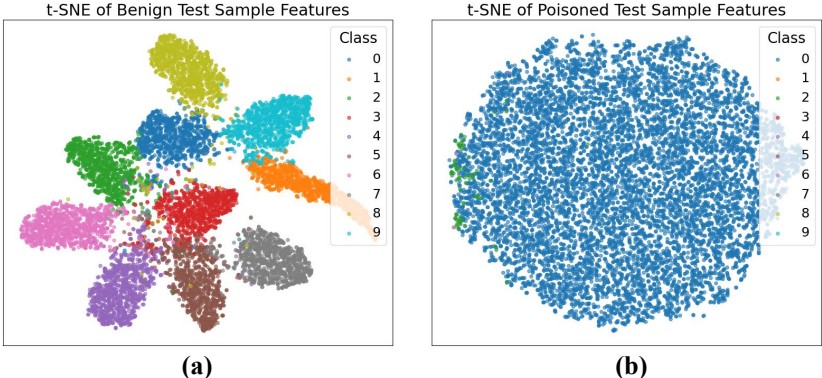

Figure 4: The t-SNE of benign and poisoned test sample features on MobileNet-V2, which is distilled from the teacher model ResNet-50 compromised by SCAR.

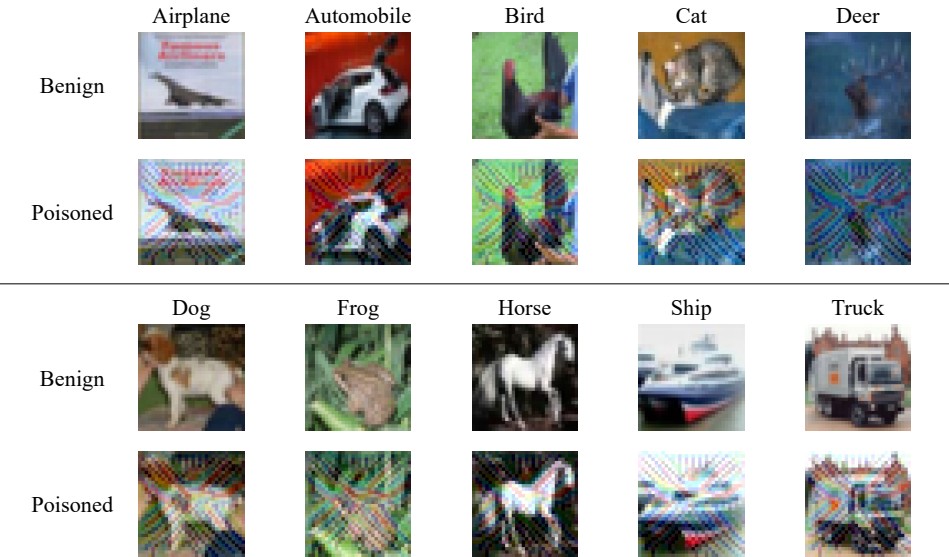

Figure 5: Examples of benign and corresponding poisoned images from the CIFAR-10 testset.

platforms but also, critically, for the student models derived through any distillation process. This dual-checking paradigm is vital for ensuring the integrity and safety of AI systems deployed in real-world, safety-critical scenarios. In essence, this paper underscores the urgent need for continuous and thorough backdoor detection throughout the AI model lifecycle from pre-deployment to post-distillation. This vigilance is crucial for guarding against evolving backdoor attacks, including various forms of conditional backdoors, and fostering trust and security in AI applications.

## K    Potential Limitations and Future Directions

Firstly, as discussion in the main results Section 4.2, the performance of SCAR on the ImageNet dataset is less satisfactory. Specifically, while SCAR consistently achieves over 80% ASR against student models in most cases on CIFAR-10, it only maintains ASR above 50% on ImageNet. This performance degradation may be attributed to the increased number of classes and larger image size in ImageNet, which intensifies the instability of the bilevel optimization process and may lead to convergence to a local minimum. Nevertheless, considering that this is the first work to reveal a new attack paradigm, we believe that the experimental results are acceptable to a certain extent. In particular, such results cannot undermine the central opinion of this paper: developers should remain vigilant and conduct thorough backdoor detection on distilled student models, even when the teacher

model and distillation dataset have been verified as secure. In future work, we aim to improve the attack effectiveness of SCAR on datasets with larger scales and greater category diversity.

Secondly, in our implicit differentiation algorithm, estimating the teacher model's gradient via the fixed-point iteration method requires synchronized linear loops, which increase the overall optimization time to some extent. We have discussed the computational overhead of our method in Appendix G and proposed potential solutions. Improving the speed and accuracy of gradient estimation remains an important direction for future research.

Finally, this paper primarily focuses on the attack threat posed by distillation-based backdoors for image classification models. Given that existing studies have shown the potential for backdoor threats to be transferred during the distillation of large language models [7], we will investigate in future work whether similar security risks exist in other modalities and tasks.

## L  Discussion on Adopted Data

In our experiments, we use only open-source datasets, namely CIFAR-10 [40], ImageNet [14] and CINIC-10 [12], for evaluation. Our research strictly adheres to the open-source licenses of these datasets and does not raise any privacy concerns. Although the ImageNet dataset may contain personal elements such as images of human faces, our work treats all data uniformly and does not intentionally exploit or manipulate any sensitive content. Therefore, our use of these datasets is fully compliant with their terms of use and does not constitute a violation of personal privacy.

## M  Ethics Statement

In this paper, we reveal a new security threat (*i.e.*, distillation-conditional backdoor attack) in the knowledge distillation process. In general, our work intends to alert developers to notice a potential false sense of security or consensus that distilling a student model based on a 'backdoor-free' teacher model with benign samples is always safe. Accordingly, our paper has positive societal impacts in general. However, we notice that the adversaries may adopt our method to execute an attack in practice. We argue that merely detecting backdoors in the teacher model is insufficient, and rigorous detection of the student model is also essential. Besides, victim developers can mitigate this threat from the source by using only trusted third-party models.

## N  Reproducibility Statement

Detailed information regarding our implementations and experiments is provided in Appendix F. Additionally, we provide the official implementation of SCAR at `https://github.com/WhitolfChen/SCAR`.

