# OpenReview forum: "Taught Well Learned Ill: Towards Distillation-conditional Backdoor Attack"
_NeurIPS.cc/2025/Conference — NeurIPS 2025 poster_

### Official Review · Reviewer_XiRf · 2025-06-21

**Clarity:** 3
**Significance:** 2
**Originality:** 2
**Rating:** 5
**Confidence:** 5

**Summary:**

This paper investigates a novel security threat in the knowledge distillation (KD) framework. Specifically, the paper introduced SCAR, a new backdoor attack method that embeds dormant backdoors into teacher models so that these backdoors remain undetectable during standard verification procedures but can be activated in student models after distillation. The SCAR formulates the problem as a bilevel optimization problem and employs an implicit differentiation algorithm with a pre-optimized trigger injection function to efficiently realize the backdoor.  Extensive experiments across various datasets, model architectures, and distillation techniques demonstrate that SCAR is highly effective and stealthy, capable of evading existing detection methods.

**Questions:**

1. Is the backdoor in the teacher model still preserved when the user fine-tunes the poisoned teacher model or uses the fine-tuning-based defense methods (such as fine-pruning)?

2. Combined with Question 1, when the defender has more attack knowledge(such as the specific attack method and parameters), can adaptive defense be designed to protect the student model?

3. I wonder about the robustness of such carefully optimized triggers. Will adding some noise to poisoned samples cause the triggers to fail? Please refer to the defense method Sampdetox [3].

Reference:

[3] Yang et al. SampDetox: Black-box Backdoor Defense via Perturbation-based Sample Detoxification. NeurIPS 2024.

**Ethical Concerns:**

["NO or VERY MINOR ethics concerns only"]

**Final Justification:**

The authors' rebuttal has addressed most of my concerns. I raise my score to 5.

**Limitations:**

Yes.

**Paper Formatting Concerns:**

No major formatting issues.

**Quality:**

3

**Strengths And Weaknesses:**

Strengths:

1. This paper uncovers a new security threat in the knowledge distillation framework, where backdoors can be hidden in teacher models while activated in student models, highlighting an underexplored area in AI security.

2. This paper conducted comprehensive experiments across diverse datasets, model architectures, and knowledge distillation techniques.

3. The paper is well written and clearly structured.

Weaknesses:

1. In ablation studies, I think it is not enough to use a fixed white patch to replace the optimized triggers. More types of triggers are worth considering. For example, since the optimized triggers cover the entire image instead of being located in corners, it is better to conduct experiments with this type of trigger.

2. It is better to conduct more experiments to investigate the resistance of SCAR to SOTA backdoor detection methods, such as MDTD [1] and TED [2].

Reference:

[1] Rajabi et al.MDTD: A Multi-Domain Trojan Detector for Deep Neural Networks. CCS 23.

[2] Mo et al. Robust Backdoor Detection for Deep Learning via Topological Evolution Dynamics. SP 2024.

---

> ### Author Rebuttal · Authors · 2025-07-31
>
> Thank you for your careful review and thoughtful comments! We are encouraged by your positive comments on our **novel security threat**, **comprehensive experiments**, **well written and clearly structured paper**, **good quality** and **good clarity**. We hope the following responses can help clarify potential misunderstandings and alleviate your concerns.
>
> ---
>
> **W1**: In ablation studies, more types of triggers are worth considering, especially the optimized triggers cover the entire image.
>
> **R1**: Thank you for this constructive suggestion! We fully agree with your point and conduct additional experiments accordingly.
> - We replace the pre-trained triggers in SCAR with image-size poisoning strategies from WaNet and BppAttack. We then train teacher models ResNet-50 on CIFAR-10 using these methods. ResNet-18 is used as the surrogate, and MobileNet-V2 serves as the student. We evaluate the attack success rates on students obtained via different KD methods.
> - As shown in Table 1, **using these image-size poisoning strategies still fails to effectively transfer backdoors to the student models** (student ASR < 25\%). It indicates that simply using image-size triggers cannot be easily 'entangled' with benign features, even if they cover the entire image, and thus fail to transfer during distillation.
>
> **Table 1**. SCAR attack performance (\%) on the student MobileNet-V2 (dubbed 'S') distilled from the teacher ResNet-50 (dubbed 'T') **under different trigger types**, using three KD methods.
> |Trigger Type|T-ACC|T-ASR|S-ACC (Response)|S-ASR (Response)|S-ACC (Feature)|S-ASR (Feature)|S-ACC (Relation)|S-ASR (Relation)|
> |-|-|-|-|-|-|-|-|-|
> |BadNets|93.81|0.72|91.47|1.03|91.49|1.48|91.22|1.38|
> |WaNet|93.07|2.20|91.07|20.97|90.47|9.73|90.77|21.22|
> |BppAttack|93.65|1.11|91.43|1.17|89.29|1.86|91.36|1.07|
> |**Ours**|92.47|1.50|91.62|99.94|91.01|99.90|91.29|99.93|
>
> We will include more discussions in the revision.
>
> ---
>
> **W2**: It is better to conduct more experiments to investigate the resistance of SCAR to SOTA backdoor detection methods, such as MDTD and TED.
>
> **R2**: Thank you for this constructive suggestion!
> - We respectfully note that we have conducted evaluation under both classical and advanced detection methods (e.g., BTI-DBF in Appendix C).
> - To further alleviate your concerns, we evaluate the suggested detection methods MDTD and TED on a ResNet-50 teacher trained on CIFAR-10 (target label 0) over 5 independent trials. Since both methods are input-level detection, we use the poisoned test set as input and compute the F1-score for poisoned sample detection. As shown in Table 2, **both methods yield very low F1-scores (< 0.06), indicating that they misclassify most poisoned samples as benign and thus failed to detect the poisoning effectively.**
>
> **Table 2**. F1-scores of two different input-level detection methods on poisoned test samples for the SCAR-attacked ResNet-50 teacher model.
> |Detection Method|1|2|3|4|5|
> |-|-|-|-|-|-|
> |MDTD|0.0488|0.0417|0.0225|0.0512|0.0580|
> |TED|0.0386|0.0281|0.0056|0.0120|0.0196|
>
> We will include more details in the revision.
>
> ---
>
> **Q1**: Does the backdoor in the teacher model persist after user fine-tuning or applying fine-tuning-based defenses (e.g., fine-pruning)?
>
> **R3**: Thank you for pointing it out! We are deeply sorry for any potential misunderstanding. We would like to provide further clarification here.
> - We respectfully note that, in our threat model, the attacker uploads a model containing a distillation-conditional backdoor to a trusted third-party platform. This platform performs backdoor detection and, upon verifying no abnormal behavior, releases the model. A victim user then downloads the model and uses it for further KD.
>     - A key aspect of this scenario is that **the platform is usually only responsible for detection, not mitigation** (like fine-tuning/pruning). Typically, uploaded models may be trained on large-scale datasets over long periods, while model developers usually only update their model instead of also updating their training data and configurations. As such, **such mitigation is generally infeasible or at least significantly degrades model performance and requires many computational resources**. Given the platform's limited resources, we assume it does not conduct non-detection defenses.
> - However, we do fully understand your concern. In practice, users can apply certain backdoor mitigation to the downloaded teacher model before performing distillation.
>     - To further alleviate your concerns, we evaluate two non-detection defenses suggested by you and another reviewer: fine-pruning and NAD. We apply these defenses to a SCAR-attacked ResNet-50 model on CIFAR-10, and then use the defended model for distillation to train MobileNet-V2.
>     - As shown in Table 3, **SCAR remains highly effective, even after the teacher model has undergone these backdoor mitigation.** We also observe that applying non-detection defenses to the teacher model may even slightly increase the attack success rate. This might be because such defenses disrupt the carefully crafted "mask" associated with the distillation-conditional backdoor, thus re-exposing the hidden backdoor.
>     - While NAD and fine-pruning do not successfully eliminate SCAR, we do not intend to claim that it is undefeatable. Rather, **our goal is to raise awareness that even models which pass backdoor detection should always not be assumed safe for distillation without further examination or cleansing.**
>
> **Table 3**. SCAR attack performance (\%) on the student MobileNet-V2 (dubbed 'S') distilled from the teacher ResNet-50 (dubbed  'T') **under two backdoor mitigation defenses**, using three KD methods.
> |Defense|T-ACC (Before Defense)|T-ASR (Before Defense)|T-ACC (After Defense)|T-ASR (After Defense)|S-ACC (Resp.)|S-ASR (Resp.)|S-ACC (Feat.)|S-ASR (Feat.)|S-ACC (Rel.)|S-ASR (Rel.)|
> |-|-|-|-|-|-|-|-|-|-|-|
> |No Defense|92.47|1.50|-|-|91.62|99.94|91.01|99.90|91.29|99.93|
> |NAD|92.47|1.50|88.45| 9.60|89.08|99.94|86.12|89.20|87.11|99.58|
> |fine-pruning|92.47|1.50|92.72|5.94|92.17|99.47|90.44|97.41|91.08|99.34|
>
> We will include more discussions in the revision.
>
> ---
>
> **Q2**: If the defender has more attack knowledge (e.g., specific attack methods and parameters), can adaptive defense be designed to protect the student model?
>
> **R4**: Thank you for this insightful question!
> - We first respectfully note that, as we mentioned in **R3**, both the platform and the end user only have access to the compromised model uploaded by the attacker **without further knowledge of the exact training data, procedures, or hyperparameters** in our threat model.
> - In particular, we do not claim the attack is undefeatable. Rather, **we encourage users to always inspect student models for backdoors**, even when both the teacher and dataset are deemed secure (see Lines 70, 83, and 334).
> - Besides, as the first work to propose distillation-conditional backdoors, we do not design additional modules to enhance the resistance to student-side defenses. **Our goal is to clearly present this novel attack surface and demonstrate a simple yet effective implementation, without introducing additional modules that may potentially obscure the threat or cause confusion.**
>
> We truly appreciate your question, which raises an important point. While it somewhat goes beyond the scope of this work, we believe it points to a promising direction for future research and will explicitly highlight it in the revision.
>
>
> ---
>
> **Q3**: Will adding some noise to poisoned samples cause the triggers to fail? Please refer to the defense method Sampdetox.
>
> **R5**: Thank you for this insightful question!
> - We hereby first clarify some potential misunderstandings.
>     - As we explained in **R3**, in our threat model, users may assume that distilling a "clean" teacher model with a benign dataset will result in a "safe" student model. However, our work shows that this assumption does not always hold: **the student model can still inherit backdoor vulnerabilities during this seemingly “secure” process.**
>     - The primary objective of our paper is to expose this novel security threat and to raise awareness. We do not intend to claim that this attack is unbreakable. Rather, **we aim to encourage users to carefully inspect student models for potential backdoors, even if the teacher model and dataset have been deemed secure** (as stated in Lines 70, 83, and 334 of the paper).
>     - As the first work to introduce distillation-conditional backdoors, we do not explore designing variants of SCAR that attempt to evade existing student-side defenses. **Our goal is to present a clear and focused analysis of this novel threat, minimizing the risk of misinterpretation.**
> - To further alleviate your concerns, we evaluate our method under your suggested Sampdetox method. Specifically, we launch the SCAR attack on a ResNet-50 teacher model on CIFAR-10, and distill three MobileNet-V2 student models using different KD methods. We then evaluate the defense performance of SampDetox using both benign and poisoned test sets.
>     - As shown in Table 4, although SampDetox can partially reduce the attack success rate of SCAR, it also leads to a noticeable drop in benign accuracy (BA). **This suggests that SCAR has already exhibited a certain degree of robustness against input pre-processing defenses**, although we have no particular design for it.
>
> **Table 4**. The defense performance (\%) of SampDetox against MobileNet-V2 student models distilled **using three different KD methods**.
> | Distillation Method | BA (Before Defense) | ASR (Before Defense) | BA (After Defense) | ASR (After Defense)|
> |-|-|-|-|-|
> | Response |91.62|99.94|84.13|56.42|
> | Feature |91.01|99.90|82.26|52.60|
> | Relation |91.29|99.93|83.65|76.01|
>
> We will include more discussions in the revision.

---

> > ### Comment · Reviewer_XiRf · 2025-08-04
> >
> > Thanks for the clarification and additional evaluation. The authors' rebuttal has addressed most of my concerns. I will raise my score to 5.

---

> > > ### Author Response · Authors · 2025-08-04
> > >
> > > Thank you for your prompt response and positive feedback! It encourages us a lot.

---

### Official Review · Reviewer_pHSe · 2025-06-27

**Clarity:** 3
**Significance:** 3
**Originality:** 3
**Rating:** 4
**Confidence:** 4

**Summary:**

This paper focuses on the potential threat of backdoor attacks on a student model distilled from a benign teacher model. By proposed a method named SCAR, the authors demonstrated that a fine-tuned teacher model may not be detected as backdoored, while a student model distilled from it may behave in a malicious way when a backdoor trigger is present. As a result, the authors suggest that a distilled model should always be examined for the backdoor risk.

**Questions:**

Please see the Concerns section above.

**Ethical Concerns:**

["NO or VERY MINOR ethics concerns only"]

**Final Justification:**

Authors have addressed my concerns, particularly on the effectiveness of SCAR with a surrogate student model. I am therefore leaning to accpentance.

**Limitations:**

Yes

**Quality:**

3

**Strengths And Weaknesses:**

**Strengths:**

1. The motivation and main message are clearly explained and easy to follow. Figure 1 is quite helpful.

2. The topic of backdoor attack for distilled models is important and seemingly less stuided.

3. I appreciate additional details in the appendix, especially on the plausible explanation for the success of SCAR.



**Concerns:**

1. The motivation of SCAR is pretty simple. It basically mimics the behavior of the student and hope to find an effective trigger after distillation. I have two concerns about the formulation of optimization objective (2):

   (i) There are a lot of hyper-parameters such as $\alpha, beta, \gamma, \delta$. While the authors provide some ablation on the choice of $\delta$, are there any insights on other parameters? Any guidelines for practioners to choose them in general?

   (ii) The effectiveness of SCAR seems to be highly dependent on the quality of this surrogate student model. As a result, what if the KD process is significantly different from the one used in Eq. (2)? For example, when the training dataset of KD is different from $\mathcal{D}$. Can authors quantify the effectiveness/robustness of the proposed method given a mismatched surrogate model?

2. The optimization process. The authors proposed to optimize the bi-level objective by directly approximating the gradient for the outer loop. This approach suffers a couple problems, e.g.,

   (i) High computation cost. This is supported by Appendix F, largely due to the calculation of Hessian and inverse Hessian.

   (ii) Non-convexity of the objective. While convexity is often assumed for the convenience of theoretical analysis, it is not the case for the problems studied in this paper. Nevertheless, I can understand that Eq. (4) is more or less motivated by this.

   I would suggest the authors to consider other optimization approches to optimize this bi-level objective. There has been a huge literature on that, such as alternative gradient descent.

3. A minor concern is the quality of backdoor triggers. Although the student is backdoored by SCAR, it looks like the poisoned image can be easily detected as in Appendix G. Can SOTA detection method successfully detect a backdoored student model or poisoned inputs for the student model?

---

> ### Author Rebuttal · Authors · 2025-07-31
>
> Thank you for your careful review and thoughtful comments! We are encouraged by your positive comments on our **clear explanation and expression**, **important research topic**, **additional details in the appendix**, **good quality**, **good clarity**, **good significance** and **good originality**. We hope the following responses can help clarify potential misunderstandings and alleviate your concerns.
>
> ---
>
> **Q1-1**: Many hyperparameters (e.g., $\alpha$, $\beta$, $\gamma$) are involved—are there insights or general guidelines for tuning them?
>
> **R1-1**: Thank you for pointing it out! We apologize for not discussing the choice of the hyperparameters $\alpha$, $\beta$, and $\gamma$ in the original paper. We would like to provide further clarification.
> - In the outer optimization objective in Eq. (2), **the scales of the four loss terms are approximately comparable** and relate to four different aspects (e.g., the student's performance on benign samples). Attackers can adjust their values based on specific needs or priorities.
> - In our experiments, we set $\alpha = \beta = \gamma = 1$ for simplicity and clarity, as this is the most straightforward configuration.
> - We fully understand your concern, and conduct ablation studies to evaluate SCAR’s sensitivity to these hyperparameters. Specifically, we train teachers (ResNet-50) using SCAR on CIFAR-10 with $\alpha$, $\beta$, and $\gamma$ independently set to 0.5, 1.0, and 1.5. We use ResNet-18 as the surrogate and MobileNet-V2 as the student. As shown in Table 1, **SCAR consistently achieves strong attack performance (student ASR > 98\%) across different hyperparameter settings**.
>
> **Table 1**. SCAR attack performance (\%) on the student MobileNet-V2 (dubbed 'S') distilled from the teacher ResNet-50 (dubbed 'T') **under different hyperparameter settings**, using three KD methods.
> |Hyperparameter Setting|T-ACC|T-ASR|S-ACC (Resp.)|S-ASR (Resp.)|S-ACC (Feat.)|S-ASR (Feat.)|S-ACC (Rel.)|S-ASR (Rel.)|
> |-|-|-|-|-|-|-|-|-|
> |$\alpha=0.5$|92.43|1.77|91.67|99.81|91.12|99.36|91.58|99.94|
> |$\alpha=1$|92.47|1.50|91.62|99.94|91.01|99.90|91.29|99.93|
> |$\alpha=1.5$|92.54|1.33|92.08|99.46|91.03|99.71|91.78|99.59|
> |$\beta=0.5$|92.21|1.61|92.36|99.37|91.30|99.04|91.33|99.07|
> |$\beta=1$|92.47|1.50|91.62|99.94|91.01|99.90|91.29|99.93|
> |$\beta=1.5$|92.35|1.43|91.99|99.83|90.87|99.01|91.52|99.36|
> |$\gamma=0.5$|92.55|1.82|92.40|99.51|91.25|98.58|91.46|99.53|
> |$\gamma=1$|92.47|1.50|91.62|99.94|91.01|99.90|91.29|99.93|
> |$\gamma=1.5$|92.13|1.88|91.74|99.97|90.98|99.77|91.64|99.23|
>
> We will include this new ablation study and discussion in the revision.
>
> ---
>
> **Q1-2**: How does SCAR perform when the KD process deviates from Eq. (2), e.g., using a KD training set different from $D$? Can the authors quantify its effectiveness or robustness given a mismatched surrogate model?
>
> **R1-2**: Thank you for these insightful questions!
> - We fully understand your concern that our effectiveness may be due to the similarties between the simulated and the real KD processes. However, we did evaluate our SCAR under various KD methods different from that used for the surrogate (in Table 1). **Our method remains effective across these settings**.
> - We argue that its effectiveness is mostly because
>   - **From the information gap view, all KD methods involve imperfect knowledge transfer, creating an information gap.** This allows hidden backdoor features to transfer while robust ones may be only partially passed (see Appendix A.1).
>   - **From the data distribution view, various KD methods still guide the student model to learn data-distribution-specific natural backdoor features.** If the distillation dataset shares the teacher’s distribution, these backdoor features tend to transfer regardless of KD strategy (see Appendix A.2).
>   - The bilevel optimization is specifically designed to embed the backdoor features into the representation learned by students. Therefore, **if the student achieves high accuracy by learning such representations—regardless of the KD method—it is likely to inherit the backdoor features to a large extent**.
> - To further alleviate your concerns, we conduct additional experiments where **the distillation dataset differs from the training set**. We train a ResNet-50 teacher using SCAR on CIFAR-10, and then distill MobileNet-V2 students using a subset of CINIC-10, which contains the same 10 classes but has a different data distribution (5,000 training and 1,000 test samples per class).
>     - As shown in Table 2, due to the distributional shift, the teacher's accuracy drops to 68.83\%. The students also experience a drop in accuracy. Although the attack success rate on the students decreased accordingly, it still remained above 54%.
>     - **These results suggest that if the student fails to achieve high accuracy during distillation—meaning it does not effectively inherit the teacher's benign knowledge—it is also less likely to inherit the backdoor.** However, such significant drops in accuracy for both teacher and student models indicates an ineffective distillation process, which is usually not meaningful in practice.
> - **We believe it is challenging to quantify the effectiveness or robustness of SCAR under a specific surrogate model, as the distillation process is inherently dynamic and variable.** Nevertheless, we would be more than happy to explore this further if you have any concrete suggestions.
>
> **Table 2**. SCAR attack performance (\%) on the student MobileNet-V2 (dubbed 'S') distilled from the teacher ResNet-50 (dubbed 'T') **with different distillation datasets**, using three KD methods.
> |Distillation Dataset|T-ACC|T-ASR|S-ACC (Response)|S-ASR (Response)|S-ACC (Feature)|S-ASR (Feature)|S-ACC (Relation)|S-ASR (Relation)|
> |-|-|-|-|-|-|-|-|-|
> |CIFAR-10|92.47|1.50|91.62|99.94|91.01|99.90|91.29|99.93|
> |CINIC-10|68.83|5.22|76.22|74.51|72.35|54.83|75.81|75.58|
>
> We will include further clarification and discussion in the revision.
>
> ---
>
> **Q2**: Optimizing the bi-level objective by directly approximating the outer-loop gradient faces issues:
> (i) High computational cost.
> (ii) Non-convexity of the objective.
> I suggest considering alternating gradient descent.
>
>
> **R2**: Thank you for this constructive suggestion!
> - **The main advantage of our derived implicit differentiation algorithm is its ability to capture the interdependence between inner and outer parameters, even when they are not part of the same model entity.** While the alternative gradient descent is indeed a commonly used optimization method, it is not designed to handle gradient propagation across *different entities* (e.g., from student to teacher), as discussed in line 194 of the paper.
> - Implicit differentiation is a classical, well-established method for bilevel optimization. Since our work is the first to propose the distillation-conditional backdoor attack, **our goal is not to over-complicate the design, but to clearly demonstrate this novel threat with a simple yet effective implementation**.
> - For (i): As we mentioned in Appendix F, our optimization procedure does introduce a non-negligible time cost. Arguably, we do not view this as a major limitation. **The optimization is performed offline and only once by the attacker, making it a one-time cost.**
> - For (ii): We fully understand your concern, and would like to provide additional clarification.
>   - **Our implicit differentiation algorithm is agnostic to the convexity of the inner problem** to some extent: as long as the inner optimization reaches a *suboptimal* solution, it suffices to estimate the outer gradient within a controllable error bound [1]. We will revise the paper to include and clarify this explanation.
>   - The KD loss used in our inner loop is essentially a combination of a log-sum-exp function (a well-known convex function [2]) and a linear function. **The sum of a convex and a linear function remains convex, which supports the validity of our approach in this context.**
>
> Thank you again for raising this point. We will emphasize the theoretical justification more clearly in the revision.
>
> **References**
>
> [1] On the iteration complexity of hypergradient computation.
> [2] Convex optimization.
>
> ---
>
> **Q3**: Poisoned images seem easily detected. Can SOTA methods reliably detect the backdoored student or its poisoned inputs?
>
> **R3**: Thank you for this insightful comment! We apologize for any potential misunderstanding. We would like to offer further clarification.
> - Arguably, **compared to traditional dataset-poisoning backdoor attacks, the visibility of the trigger is less critical in our threat model.** This is because the poisoned training samples are not accessible to either the platform or the end users, and only take effect at inference time.
> - You may further worry about why we don't conduct more detection on student models.
>   - In our threat model, an adversary uploads a teacher model with a distillation-conditional backdoor to a verification platform. The platform fails to detect the dormant backdoor and marks the model as benign. A victim user then downloads it and performs KD using a clean dataset. **This scenario is realistic and meaningful, as users often assume that distilling a verified "clean" model with a benign dataset yields a "safe" student model.** However, our work shows that the student model can still inherit backdoor vulnerabilities during this seemingly “secure” process.
>   - Our goal is to reveal this novel threat and raise awareness. We do not claim the attack is undetectable or undefeatable. Rather, **we encourage users to always inspect student models for backdoors**, even when both the teacher and dataset are deemed secure (see Lines 70, 83, and 334).
>
> Nevertheless, we fully agree that enhancing the student-side evasiveness and the trigger's stealthiness is an important and meaningful direction. We will add more discussions in the revision.

---

> > ### Comment · Reviewer_pHSe · 2025-08-04
> >
> > I thank the authors for the response and additional experiments. I am still confused about the insight behind the proposed method. The information gap does not explain why backdoor triggers can be preseved through a different KD process; regarding data distribution view, if a student learns the behavior of a teacher nearly perfectly, the backdoor pattern will not lead to a malicious behavior as the teacher responds normally when such backdoor pattern exists in the input.

---

> ### Author Response · Authors · 2025-08-04
> **Thank you and our responses to your post-rebuttal comments.**
>
> Thank you for your prompt response and insightful feedback! We sincerely appreciate your engagement. We would, however, like to respectfully clarify a possible misunderstanding concerning these two perspectives.
> - We first have to acknowledge that rigorously establishing a theoretical foundation for the effectiveness of SCAR is inherently challenging, given the complexity of deep neural networks. The analyses provided in this paper are intended as a preliminary effort to shed light on its underlying mechanisms.
> - From the perspective of the information gap, we intended to suggest that **SCAR may exploit the *inherent* discrepancies arising from imperfect knowledge transfer during knowledge distillation (KD) to activate backdoors in the student model**. We respectfully note that, perhaps counterintuitively, **such information gaps are intrinsic to the KD process**, as **the student model is unlikely to precisely replicate the behavior of the teacher model**. This divergence may stem from architectural differences, limited model capacity, or the inherently lossy nature of the distillation objective [1][2]. Consequently, while the teacher model may behave benignly, the student model can still manifest backdoor behaviors.
> - From the data distribution perspective, we intended to suggest that **SCAR may leverage distribution-specific natural backdoor features to facilitate the persistence of backdoors across different KD processes**. Regardless of the KD method employed, the primary objective is to optimize the student model's performance on the given data distribution. However, during this process, the student model distilled from datasets drawn from the same distribution as the teacher may also implicitly learn certain distribution-specific natural backdoor features. This might allow the backdoor trigger to persist across different KD strategies.
>
> We sincerely appreciate your insightful feedback. We hope this additional clarification helps address your concerns. We plan to further investigate distillation-conditioned backdoor threats and their underlying mechanisms in future work.
>
> **References**:
>
> [1] On the efficacy of knowledge distillation.
> [2] Knowledge distillation: A survey.

---

> > ### Comment · Reviewer_pHSe · 2025-08-06
> >
> > I appreciate authors' further clarification, which addressed my concerns.

---

> > > ### Author Response · Authors · 2025-08-06
> > >
> > > Thank you for your prompt response and positive feedback! It encourages us a lot. We hope the additional evidence and clarifications warrant your favorable reassessment.

---

### Official Review · Reviewer_8GaK · 2025-06-30

**Clarity:** 2
**Significance:** 3
**Originality:** 3
**Rating:** 4
**Confidence:** 4

**Summary:**

The paper proposed a backdoor defense approach specifically for the knowledge-distillation setting using a bilevel optimization method. The approach follows a workflow of distributor -> verifier -> user, where the teacher model first needs to pass backdoor detection defenses. The outer optimization takes into account the "static" part of KD-resistant backdoor injection while the inner optimization handles the "dynamic" KD process. The pipeline is done by training a surrogate model so no prior knowledge on user-end architecture needed. Additionally, the paper proposed to overcome the challenging optimization with an implicit differentiation algorithm with a pre-optimized trigger injection function. Under the threat model that the attacker has the access and the freedom to inject backdoor to the teacher model, the paper showed that their approach is capable of beating ADBA with a huge margin especially when 1. the datasets are more challenging like ImageNet and 2. the architectural differences between the teacher and the student models are more different.

**Questions:**

Q1. Can the authors further elaborate on the inner optimization in (2)? Does the KD loss depend on the specific kind of KD used? If not, it is not clear to me how minimizing the KL divergence between the output logits of F_t and F_s (which seems to be Response-based KD) can improve backdoor resistance against other KD strategies.

Q2. I think it would be interesting to see the L2 loss pattern of backdoor vs clean samples during the training of model, and how well the approach can bypass the verifier if we replace detection-based method with other kinds of defenses such as ABL [1], which also has a backdoor detection component, assuming the verifier has a small subset of the training set?

Q3. How does the approach work against SOTA distillation-based defenses like NAD?

Q4. Does the choice of the surrogate model matter? For instance, does backdoor generation using a smaller/bigger surrogate model transfer better in KD?

S1. The detection methods used in the experiments are overall too simple. I wonder how well this approach can bypass more advanced detection methods like A2D [2] or BAN [3].

[1] Li, Yige, Xixiang Lyu, Nodens Koren, Lingjuan Lyu, Bo Li, and Xingjun Ma. "Anti-backdoor learning: Training clean models on poisoned data." Advances in Neural Information Processing Systems 34 (2021): 14900-14912.
[2] Samar Fares and Karthik Nandakumar. Attack to defend: Exploiting adversarial attacks for detecting poi-
soned models.
[3] Xiaoyun Xu, Zhuoran Liu, Stefanos Koffas, Shujian Yu, and Stjepan Picek. Ban: Detecting backdoors
activated by adversarial neuron noise. Advances in Neural Information Processing Systems, 37:114348–
114373, 2025.

**Ethical Concerns:**

["NO or VERY MINOR ethics concerns only"]

**Final Justification:**

The authors addressed all my concerns and I gave an acceptance score already. Given the limitations of scope of this work, I would like to keep my score as borderline accept.

**Limitations:**

yes

**Quality:**

3

**Strengths And Weaknesses:**

Strengths -
* Overall the paper is well-written and technically sounding.
* The approach seems to be novel.
* The authors provided theoretical justifications of their approach.
* The threat model assumed is realistic and practical

Weaknesses -
* Limited comparisons against prior works, but understandable since KD-resistant backdoor attacks are relatively new
* Weak detection methods used in experiments
* No experiments against existing non-detection defenses

---

> ### Author Rebuttal · Authors · 2025-07-31
>
> Thank you for your careful review and thoughtful comments! We are encouraged by your positive comments on our **well-written and technically sounding paper**, **novel approach**, **theoretical justifications of our approach**, **realistic and practical threat model**, **good quality**, **good significance** and **good originality**. We hope the following responses can help clarify potential misunderstandings and alleviate your concerns.
>
> ---
>
> **W1**: Limited comparisons against prior works, but understandable since KD-resistant backdoor attacks are relatively new.
>
> **R1**: Thank you for pointing it out! We sincerely apologize for any misunderstanding. We hereby provide further clarification.
> - The threat proposed in our paper is **distillation-conditional backdoor (DCB) rather than distillation-resistant backdoor (DRB).**
>   - DRB aims to transfer **active** backdoors from the teacher to the student during distillation.
>   - DCB aims to implant a **dormant** backdoor into the teacher that remains inactive and **only becomes activated after the student is distilled**. The teacher must appear entirely benign, while the student exhibits malicious behavior. This makes DCB more challenging and, arguably, more threatening than DRB.
> - As the first work to propose DCB, we can only compare against adapted variants of related methods. Section 2.2 of our paper discusses DRB, which remains a relatively new and underexplored area. **We adapted the most representative method in this domain (ADBA) as our baseline**.
>
> We will add more discussions in our revision.
>
> ---
>
> **W2&S1**: Weak detection methods used in experiments. I wonder how well this approach can bypass more advanced detection methods like A2D or BAN.
>
> **R2**: Thank you for these constructive suggestions!
> - We respectfully note that we have conducted evaluation under both classical and advanced detection methods (e.g., BTI-DBF in Appendix C).
> - To further alleviate your concerns, we conduct additional evaluation using your suggested detection methods. Specifically, we conduct 5 independent detection trials on a ResNet-50 teacher trained on CIFAR-10 with target label 0.
>   - For A2D, following the sensitivity to adversarial perturbations (SAP) protocol, a backdoored model is expected to show high SAP values. As shown in Table 1, across 5 trials, **the SCAR-attacked teacher consistently exhibits low SAP, indicating that A2D failed to detect the backdoor**.
>   - For BAN, we record both the prediction distribution after mask training and the predicted target class. As shown in Table 2, across all 5 trials, **BAN also consistently misjudges the backdoor target class, leading to a high false positive rate**.
>
> **Table 1**. SAP values reported by A2D.
> ||1|2|3|4|5|
> |-|-|-|-|-|-|
> |SAP|0.117|0.117|0.185|0.206|0.027|
>
> **Table 2**. Prediction distributions generated by BAN. **Bolded**: the target labels inferred by BAN.
> |Class $\rightarrow$|0|1|2|3|4|5|6|7|8|9|
> |-|-|-|-|-|-|-|-|-|-|-|
> |1|434|461|**1943**|459|43|558|151|46|454|451|
> |2|333|**2628**|272|70|146|38|0|253|766|494|
> |3|141|432|260|1028|11|6|111|**2057**|536|418|
> |4|297|43|215|266|**3066**|287|228|89|285|224|
> |5|176|371|696|79|65|1182|123|**1512**|209|587|
>
> We will add more details in our revision.
>
> ---
>
> **W3&Q3**: No experiments against existing non-detection defenses. How does the approach work against SOTA defenses like NAD?
>
> **R3**: Thank you for pointing it out! We are deeply sorry for any potential misunderstanding and hereby offer further clarification.
> - We respectfully note that, in our threat model, the attacker uploads a model containing a DCB to a trusted third-party platform. This platform performs backdoor detection and, upon verifying no abnormal behavior, releases the model. A victim user then downloads the model and uses it for further KD.
>   - A key aspect of this scenario is that **the platform is usually only responsible for detection, not mitigation**. Typically, uploaded models may be trained on large-scale datasets over long periods, while developers usually only upload their model without uploading training data and configurations. As such, **such mitigation is generally infeasible or at least significantly degrades model performance and requires many computational resources**. Given the platform's limited resources, we assume it does not conduct non-detection defenses.
> - We fully understand your concern, where users may apply mitigation methods to the downloaded teacher model before KD.
>     - To further alleviate your concerns, we evaluate two non-detection defenses suggested by you and another reviewer: NAD and fine-pruning. We apply these defenses to a SCAR-attacked ResNet-50 on CIFAR-10, and then use the defended model for KD to train MobileNet-V2.
>     - As shown in Table 3, **SCAR remains highly effective, even after the teacher has undergone these backdoor mitigation.** Besides, using non-detection defenses may even slightly raise ASR. This might be because such defenses disrupt the carefully crafted "mask" of the DCB, thus re-exposing the hidden backdoor.
>     - While NAD and fine-pruning fail to eliminate SCAR, we don't intend to claim that it is undefeatable. Rather, **our goal is to raise awareness that even models which pass backdoor detection should always not be assumed safe for distillation without further examination or cleansing.**
> - Besides, as the first work to propose DCB, we intentionally avoid incorporating complex techniques to bypass potential defenses. **We aim to clearly define the threat and present a simple, effective instantiation (SCAR), avoiding unnecessary design complexity that could cause confusion.**
>
> **Table 3**. SCAR attack performance (\%) on the student MobileNet-V2 (dubbed 'S') distilled from the teacher ResNet-50 (dubbed 'T') under two mitigation methods, using three KD methods.
> |Defense|T-ACC (w/o Defense)|T-ASR (w/o Defense)|T-ACC (w/ Defense)|T-ASR (w/ Defense)|S-ACC (Resp.)|S-ASR (Resp.)|S-ACC (Feat.)|S-ASR (Feat.)|S-ACC (Rel.)|S-ASR (Rel.)|
> |-|-|-|-|-|-|-|-|-|-|-|
> |No Defense|92.47|1.50|-|-|91.62|99.94|91.01|99.90|91.29|99.93|
> |NAD|92.47|1.50|88.45| 9.60|89.08|99.94|86.12|89.20|87.11|99.58|
> |fine-pruning|92.47|1.50|92.72|5.94|92.17|99.47|90.44|97.41|91.08|99.34|
>
> We will include more discussions in the revision.
>
> ---
>
> **Q1**: Can the authors elaborate on the inner optimization in (2)? Does the KD loss vary with the KD method used?
>
> **R4**: Thank you for these insightful questions!
> - In (2), the inner optimization **adopts the standard knowledge distillation (KD) loss** to guide the surrogate to approximate the knowledge transfer process during distillation. Specifically, we minimize the KL divergence between the logits of the teacher and the surrogate through a finite number of gradient steps.
> - We fully understand your concern that our effectiveness may be due to the similarties between the simulated and the real KD processes. However, we did evaluate our SCAR under various KD methods different from that used for the surrogate (in Table 1). **Our method remains effective across these settings**.
> - We argue that its effectiveness is mostly because
>   - **From the information gap view, all KD methods involve imperfect knowledge transfer, creating an information gap.** This allows hidden backdoor features to transfer while robust ones may be only partially passed (see Appendix A.1).
>   - **From the data distribution view, various KD methods still guide the student to learn implicit, data-distribution-specific natural backdoor features.** If the distillation dataset shares the teacher’s data distribution, these backdoor features tend to transfer regardless of KD strategy (see Appendix A.2).
>   - The bilevel optimization in SCAR is specifically designed to embed the backdoor features into the representation learned by the students. Therefore, **if the student achieves high accuracy by learning such representations—regardless of the KD method—it is also likely to inherit the backdoor features to a large extent**.
>
> We will further clarify these points in the revision to help readers better understand the underlying reasons for SCAR’s effectiveness.
>
> ---
>
> **Q2**: I’m interested in the L2 loss patterns of backdoor vs. clean samples during training and how well the method bypasses verifiers using defenses like ABL, assuming the verifier holds a small training subset.
>
> **R5**: Thank you for this insightful question!
> - As we mentioned in **R3**, in our attack scenario, the attacker only uploads a compromised (potentially poisoned) model to the platform (verifier), but does not upload the corresponding training dataset.
> - Even if we assume that the attacker will upload parts of the training/testing data, they will only upload the benign ones instead of also the poisoned ones.
> - When the user downloads the model and performs knowledge distillation, they only use clean samples, i.e., **no poisoned samples are involved in the distillation process**. Therefore, training-time detection methods (and their follow-up mitigation) like ABL are not feasible in our attack scenario.
>
> We will include more discussions in the revision.
>
> ---
>
> **Q4**: Does the choice of the surrogate model matter? Does backdoor generation using a smaller/bigger surrogate model transfer better in KD?
>
> **R6**: Thank you for these constructive suggestions! We are deeply sorry for any potential misunderstanding.
> - We would like to first respectfully clarify that our main experiments already evaluate settings with mismatched surrogate and student models (Table 1), where our **SCAR remains effective** under these settings.
> - To further alleviate your concerns, we evaluate SCAR on CIFAR-10 with different surrogates (ResNet-18, ResNet-34, DenseNet-121, SqueezeNet), and **it consistently achieves high attack success rates** (student ASR > 95%) across all settings (See our **R1** to **Reviewer TVht** for more details).

---

> ### Author Response · Authors · 2025-08-04
> **Thanks to Reviewer 8GaK**
>
> Please allow us to thank you again for reviewing our paper and the valuable feedback, and in particular for recognizing the strengths of our paper in terms of our well-written and technically sounding paper, novel approach, theoretical justifications of our approach, realistic and practical threat model, good quality, good significance and good originality.
>
> Please let us know if our response and the new experiments have properly addressed your concerns. We are more than happy to answer any additional questions during the post-rebuttal period. Your feedback will be greatly appreciated.

---

> > ### Comment · Reviewer_8GaK · 2025-08-05
> > **Response to authors**
> >
> > Thank you for your detailed response. I appreciate the efforts put in this rebuttal which have effectively addressed all my concerns.

---

> > > ### Author Response · Authors · 2025-08-06
> > >
> > > Thank you for your prompt response and positive feedback! It encourages us a lot. We hope the additional evidence and clarifications warrant your favorable reassessment.

---

### Official Review · Reviewer_TVht · 2025-07-02

**Clarity:** 2
**Significance:** 3
**Originality:** 3
**Rating:** 4
**Confidence:** 5

**Summary:**

This paper introduces SCAR , a novel distillation-conditional backdoor attack (DCBA) framework that exploits the KD process to implant dormant backdoors in teacher models. These backdoors remain inactive during normal inference but are activated in student models after KD, even when the distillation dataset is benign. The attack is formulated as a bilevel optimization problem: the inner loop simulates KD by optimizing a surrogate student model, while the outer loop trains the teacher model to maximize the attack success rate on poisoned samples in the student model.

**Questions:**

1. **Surrogate Model Design**:  How do architectural differences (e.g., CNN vs. ViT) between the surrogate and target student models impact SCAR’s attack success rate? For instance, would a CNN-based surrogate fail to simulate KD for a ViT student?

2. **Defense Evaluation**:  The paper claims SCAR evades detection but does not test the surrogate model against state-of-the-art detections/defenses. How effective are these methods against SCAR, and what modifications could improve stealthiness?

3. **Optimization Validity**: The bilevel optimization assumes differentiability of the inner loop (KD simulation). How sensitive is SCAR to approximations in this process (e.g., finite inner steps, surrogate initialization)?

**Ethical Concerns:**

["NO or VERY MINOR ethics concerns only"]

**Final Justification:**

The clarifications and modifications provided during the rebuttal process have resolved my issues to a satisfactory degree. As a result, I have decided to maintain my final rating as **Borderline Accept**. I appreciate the authors' efforts in engaging with the feedback and improving the manuscript.

**Limitations:**

Yes. The authors acknowledge limitations, such as scalability to larger datasets and potential failure cases.

**Quality:**

3

**Strengths And Weaknesses:**

#### **Strengths**
1. The paper introduces a novel attack paradigm (DCBA) that bridges the gap between backdoor attacks and KD. This work first explores how KD itself can activate dormant backdoors.
2. Extensive experiments on CIFAR-10, ImageNet, and multiple architectures (ResNet, VGG, ViT, MobileNet, ShuffleNet, EfficientViT) validate SCAR’s effectiveness.  Comparison with baseline ADBA(FT) shows SCAR achieves significantly higher ASR in student models while maintaining low ASR in the teacher model.

#### **Weaknesses**
1. **Limited Surrogate Model Analysis**: The paper does not thoroughly explore how structural or task mismatches between the surrogate and target student models affect attack performance. This paper also lacks analysis on more structures of surrogate models.
2. **Scalability Concerns**: The paper acknowledges that SCAR’s effectiveness may degrade on larger-scale datasets (e.g., full ImageNet), but no experiments or theoretical analysis address this limitation.
3. **Optimization Assumptions**: The bilevel optimization relies on implicit differentiation and pre-optimized triggers, which may introduce biases or instabilities if the surrogate model poorly approximates real-world KD dynamics.
4. **Stealthiness of Student**: A key concern is whether the student model can evade existing backdoor detection methods. If it cannot, the practicality of the attack may be limited, as such models would likely be discarded in real-world deployment. This aspect requires further investigation and discussion.

---

> ### Author Rebuttal · Authors · 2025-07-31
>
> Thank you for your careful review and thoughtful comments! We are encouraged by your positive comments on our **novel attack paradigm**, **extensive experiments**, **good quality**, **good significance** and **good originality**. We hope the following responses can help clarify potential misunderstandings and alleviate your concerns.
>
> ---
>
> **W1&Q1**: The paper lacks evaluation of how architectural or task mismatches between surrogate and student models affect attack performance, especially CNN surrogates for ViT students. It also omits analysis involving more diverse surrogate structures.
>
> **R1**: Thank you for these constructive suggestions! We would like to respectfully clarify that **the effectiveness of SCAR does not rely on architectural similarity between surrogate and student models**.
> - As stated in line 252, we used ResNet-18 as the surrogate, while the students are structurally different: MobileNet-V2, ShuffleNet-V2, and EfficientViT. **This represents a challenging setting**, i.e., the surrogate is highly different from the student.
>   - **Using CNN-based surrogates attacking ViT-like students is covered in our paper**, e.g., ResNet-18 (surrogate) vs. EfficientViT (student), where SCAR remains effective.
>   - SCAR's effectiveness mostly stems from two factors: the information gap introduced by KD and the shared data distribution between the training and distillation datasets (see Appendix A). Besides, the bilevel optimization is specifically designed to embed backdoor features into the representations learned by the student. Thus, **if the student achieves high accuracy by learning such representations—regardless of the KD method—it is likely to inherit the backdoor features to a large extent.**
> - To further address your concern, we evaluate SCAR on CIFAR-10 with different surrogates (ResNet-18, ResNet-34, DenseNet-121, SqueezeNet), using ResNet-50 and MobileNet-V2 as teacher and student. As shown in Table 1, **SCAR consistently achieves high attack success rates** (student ASR > 95%) across all settings.
>
> **Table 1**. SCAR attack performance (\%) with **different surrogate models** on the student MobileNet-V2 (dubbed 'S') distilled from the teacher ResNet-50 (dubbed 'T') using three KD methods.
> |Surrogate|T-ACC|T-ASR|S-ACC (Resp.)|S-ASR (Resp.)|S-ACC (Feat.)|S-ASR (Feat.)|S-ACC (Rel.)|S-ASR (Rel.)|
> |-|-|-|-|-|-|-|-|-|
> |ResNet-18|92.47|1.50|91.62|99.94|91.01|99.90|91.29|99.93|
> |ResNet-34|92.32|1.42|92.26|99.47|90.77|99.68|91.76|99.92|
> |DenseNet-121|92.79|1.43|92.19|99.80|90.67|95.78|91.80|99.90|
> |SqueezeNet|92.41|1.56|91.86|99.94|91.16|99.13|91.78|99.88|
>
> We will add more details in our revision.
>
> ---
>
> **W2**: The paper notes SCAR’s effectiveness may decline on large-scale datasets but provides no experiments or theory to support this.
>
> **R2**: Thank you for pointing this out!
> - As illustrated in Appendix I, SCAR may indeed face this limitation.
>   - This limitation is largely because large-scale datasets can intensify the instability of the bilevel optimization, potentially causing convergence to a poor local minimum.
>   - **We have also analyzed SCAR’s effectiveness through the perspectives of information gap and data distribution** (in Appendix A), which helps explain this limitation:
>     - Information Gap View: large-scale datasets make it more difficult to exploit knowledge transfer gaps during distillation for backdoor injection, significantly increasing the complexity of bilevel optimization.
>     - Data Distribution View: the increased complexity of large-scale datasets makes it harder for the student model to learn data-distribution-specific backdoor features.
> - Nevertheless, we argue this doesn't undermine the paper's core message: **users should remain vigilant and always perform backdoor detection on distilled student models**, even when both the teacher and distillation dataset have been verified as secure. Despite the lower ASR (just over 50%) on ImageNet, the threat remains significant and non-negligible.
> - As the first work introducing this novel threat, we chose not to incorporate additional modules used to overcome this potential limitation. **Our goal is to present a clean and minimal version of SCAR in order to make the nature of distillation-conditional backdoors more clear and avoid introducing potential confusion.**
>
> Nevertheless, we fully agree that enhancing attack performance on large-scale datasets is a valuable research direction. We will explore it in the future.
>
> ---
>
> **W3**: The bilevel optimization depends on implicit differentiation and pre-optimized triggers, which may cause bias or instability if the surrogate model poorly reflects real KD dynamics.
>
> **R3**: Thank you for this insightful comment!
> - We fully understand your concern that our effectiveness may due to the similarties between simulated and real KD processes. However, we did evaluate our SCAR under various KD methods different from that used for the surrogate (in Table 1). **Our method remains effective across these settings**.
> - We argue that its effectiveness is mostly because
>   - Information Gap View: **all KD methods involve imperfect knowledge transfer, creating an information gap**. This allows hidden backdoor features to transfer while robust ones may be only partially passed (see Appendix A.1).
>   - Data Distribution View: **various KD methods still guide the student model to learn distribution-specific natural backdoor features.** If the distillation dataset shares the teacher’s distribution, these backdoor features tend to transfer regardless of KD strategy (see Appendix A.2).
>   - Our bilevel optimization is specifically designed to embed the backdoor features into the representation learned by the students. Therefore, **if the student achieves high accuracy by learning such representations, regardless of the KD method, it is also likely to inherit the backdoor features to a large extent**.
>
> We will add more details in the revision.
>
> ---
>
> **W4**: A key concern is whether the student can evade backdoor detection.
>
> **R4**: Thank you for this insightful comment! We apologize for any potential misunderstanding and hereby provide further clarification.
> - In our threat model, an adversary uploads a teacher model with a distillation-conditional backdoor to a verification platform. The platform fails to detect the dormant backdoor and marks the model as benign. A victim user then downloads it and performs KD using a clean dataset. **This scenario is realistic and meaningful, as users often assume that distilling a verified "clean" model with a benign dataset yields a "safe" student model.** However, our work shows that the student model can still inherit backdoors during this seemingly “secure” process.
> - Our goal is to reveal this novel threat and raise awareness. We do not claim the attack is undetectable or undefeatable. Rather, **we encourage users to always inspect student models for backdoors**, even when both the teacher and dataset are deemed secure (see Lines 70, 83, and 334).
> - As the first to propose distillation-conditional backdoors, we focus on presenting a clear, minimal version of SCAR, without designing variants to evade student-side detection. This helps ensure clarity and avoid potential misunderstandings.
>
> We will add more discussions in the revision.
>
> ---
>
> **Q2**: The paper does not test the surrogate model against SOTA defenses. How effective are these methods, and how can SCAR improve its stealth?
>
> **R5**: Thank you for this insightful question!
> - The surrogate model is **only used by the attacker during the training of the poisoned teacher model** to simulate the user’s distillation process. It is **not exposed** to the verification platform or victim users, and thus neither detection nor defense is feasible.
> - We speculate your concern may relate to whether distilled student models can be detected. This is indeed an important question.
>   - As noted in **R4**, our threat model is realistic and highlights a key risk: **even if the teacher and dataset are clean, the student may still inherit backdoors.** Users should rigorously inspect student models.
>   - We agree that enhancing SCAR’s stealthiness against student-side detection is a promising future direction. This is something we are very interested in exploring in our future work.
>
> ---
>
> **Q3**: The bilevel optimization assumes differentiability of the inner loop. How sensitive is SCAR to approximations in this process (e.g., finite inner steps, surrogate initialization)?
>
> **R6**: Thank you for this insightful question! We hereby offer further clarification:
> - **The surrogate model is randomly re-initialized** at the start of each outer epoch; no special initialization is used.
> - We fix the number of inner optimization steps to 20 in our main experiments. **This is mostly because the surrogate model has already reached a reasonably good local optimum** (around 85% accuracy compared to the teacher) within these steps, which is sufficient for guiding the outer optimization.
> - To assess sensitivity to this setting, we conduct ablation studies using different inner steps (20, 30, 40) when attacking ResNet-50 on CIFAR-10. The surrogate is ResNet-18, and the student is MobileNet-V2. As shown in Table 2, **SCAR remains effective under all settings.** This suggests that once the inner parameters are optimized to a nearly local optimum, the outer gradient is adequately approximated.
>
> **Table 2**. SCAR attack performance (\%) with **different inner steps** on the student MobileNet-V2 (dubbed 'S') distilled from the teacher ResNet-50 (dubbed 'T') using three KD methods.
> |Inner Step|T-ACC|T-ASR|S-ACC (Resp.)|S-ASR (Resp.)|S-ACC (Feat.)|S-ASR (Feat.)|S-ACC (Rel.)|S-ASR (Rel.)|
> |-|-|-|-|-|-|-|-|-|
> |20 steps|92.47|1.50|91.62|99.94|91.01|99.90|91.29|99.93|
> |30 steps|92.35|1.46|92.13|99.41|91.03|99.04|91.26|99.39|
> |40 steps|92.20|1.51|91.98|99.69|91.16|99.18|91.87|99.49|
>
> We will include this analysis in the revision.

---

> > ### Comment · Reviewer_TVht · 2025-08-03
> >
> > Thank you for your detailed response and the additional experiments, which have effectively addressed my concerns.

---

> > > ### Author Response · Authors · 2025-08-04
> > >
> > > Thank you for your prompt response and positive feedback! It encourages us a lot. We hope the additional evidence and clarifications warrant your favorable reassessment.

---

### Note · Authors · 2025-08-12

We sincerely thank all reviewers for their insightful feedback and constructive suggestions. We are deeply encouraged by their positive evaluation of **our work's novelty, significance, technical soundness, and clarity**. We are pleased that our detailed responses and additional experiments have addressed their main concerns. This final remark summarizes our key clarifications and new results:

- **Effectiveness across Different Settings**: To address concerns regarding effectiveness, our new experiments indicate that SCAR achieves a high student ASR (>95\%) across diverse settings, including attacks **using various surrogate model architectures** such as ResNet-18, ResNet-34, DenseNet-121, and SqueezeNet. Further ablation studies show that SCAR is robust to hyperparameter choices ($\alpha, \beta, \gamma$) or the number of inner optimization steps.
- **Confirmed Stealth and Resilience**: In response to questions on resilience, our additional results suggest SCAR's strong ability to evade defenses. On the teacher model, **it successfully bypasses reviewer-suggested detection methods**, including A2D, BAN, MDTD, and TED. Furthermore, SCAR's potency persists when non-detection methods are applied, both to the teacher model (e.g., NAD, fine-pruning) and to the student model (e.g., SampDetox).
- **Clarifications on Threat Model and Broader Implications**: We further clarify our distillation-conditional backdoor threat model, representing a new class of **"Emergent Vulnerability"** in AI. Such security threats can arise in student models during KD, even if the teacher model is clean. This occurs because the student imperfectly learns from the teacher, allowing latent threats to surface—hence our title, *"Taught Well, Learned Ill"*. **Our goal is to expose these novel threats and raise awareness.** *We do not claim these threats are undetectable or undefeatable; rather, we urge users to always inspect student models, even when the KD process seems secure*.
- **Analysis of Effectiveness**: We also offer further analysis on the effectiveness of SCAR, attributing it to two key factors: 1) exploiting the inherent **information gap** during knowledge transfer, and 2) leveraging **data-distribution-specific** features that facilitate the persistence of backdoors.

We sincerely thank the reviewers again for their valuable time and effort. We believe this process has helped strengthen our paper and will incorporate all new results and discussions into the revision.

---

### Decision · Program_Chairs · 2025-09-17

**Decision:**

Accept (poster)

**Comment:**

This paper investigates the risks associated with knowledge distillation, a common practice where users train a smaller "student" model from a large, imported "teacher" model. The authors propose a method to inject a dormant backdoor into the teacher model that is only activated through this distillation process. Reviewers agreed that this attack setting is both innovative and realistic. Although initial concerns were raised about the experiments, the assumptions, and the sensitivity to hyperparameters, they were well-addressed during the rebuttal.